# Online Sampling from Log-Concave Distributions

**Holden Lee**
Duke University

**Oren Mangoubi**
Worcester Polytechnic Institute

**Nisheeth K. Vishnoi**
Yale University

## Abstract

Given a sequence of convex functions $f_0, f_1, \ldots, f_T$, we study the problem of sampling from the Gibbs distribution $\pi_t \propto e^{-\sum_{k=0}^{t} f_k}$ for each epoch $t$ in an *online* manner. Interest in this problem derives from applications in machine learning, Bayesian statistics, and optimization where, rather than obtaining all the observations at once, one constantly acquires new data, and must continuously update the distribution. Our main result is an algorithm that generates roughly independent samples from $\pi_t$ for every epoch $t$ and, under mild assumptions, makes $\mathrm{polylog}(T)$ gradient evaluations per epoch. All previous results imply a bound on the number of gradient or function evaluations which is at least linear in $T$. Motivated by real-world applications, we assume that functions are smooth, their associated distributions have a bounded second moment, and their minimizer drifts in a bounded manner, but do not assume they are strongly convex. In particular, our assumptions hold for online Bayesian logistic regression, when the data satisfy natural regularity properties, giving a sampling algorithm with updates that are poly-logarithmic in $T$. In simulations, our algorithm achieves accuracy comparable to an algorithm specialized to logistic regression. Key to our algorithm is a novel stochastic gradient Langevin dynamics Markov chain with a carefully designed variance reduction step and constant batch size. Technically, lack of strong convexity is a significant barrier to analysis and, here, our main contribution is a martingale exit time argument that shows our Markov chain remains in a ball of radius roughly poly-logarithmic in $T$ for enough time to reach within $\varepsilon$ of $\pi_t$.

## 1 Introduction

In this paper, we study the following online sampling problem:

**Problem 1.1.** *Consider a sequence of convex functions $f_0, f_1, \ldots, f_T : \mathbb{R}^d \to \mathbb{R}$ for some $T \in \mathbb{N}$, and let $\varepsilon > 0$. At each epoch $t \in \{1, \ldots, T\}$, the function $f_t$ is given to us, so that we have oracle access to the gradients of the first $t + 1$ functions $f_0, f_1, \ldots, f_t$. The goal at each epoch $t$ is to generate a sample from the distribution $\pi_t(x) \propto e^{-\sum_{k=0}^{t} f_k(x)}$ with fixed total-variation (TV) error $\varepsilon$. The samples at different time steps should be almost independent.*

Various versions of this problem have been considered in the literature, with applications in Bayesian statistics, optimization, and theoretical computer science; see [NR17, DDFMR00, ADH10] and references therein. If $f$ is convex, then a distribution $p \propto e^{-f}$ is logconcave; this captures a large class of useful distributions such as gaussian, exponential, Laplace, Dirichlet, gamma, beta, and chi-squared distributions. We give some settings where online sampling can be used:

- **Online posterior sampling.** In Bayesian statistics, the goal is to infer the probability distribution (the *posterior*) of a parameter, based on observations; however, rather than obtaining all the observations at once, one constantly acquires new data, and must continuously update the posterior distribution, rather than only after all data is collected. Suppose $\theta \sim p_0 \propto e^{-f_0}$ for a given *prior distribution*, and samples $y_t$ drawn from the conditional distribution $p(\cdot|\theta, y_1, \ldots, y_{t-1})$ arrive in a streaming manner. By Bayes's rule, letting $p_t(\theta) = e^{-f_t(\theta)} := p(\theta|y_1, \ldots, y_t)$ be the posterior distribution, we have the following recursion: $p_t(\theta) \propto p_{t-1}(\theta)p(y_t|\theta, y_1, \ldots, y_{t-1})$.

Hence, $p_t(\theta) \propto e^{-\sum_{k=0}^{t} f_k(\theta)}$. The goal is to sample from $p_t(\theta)$ for each $t$. This fits the setting of Problem 1.1 if $p_0$ and all updates $p(y_t|\theta, y_1, \ldots y_{t-1})$ are logconcave.

One practical application is online logistic regression; logistic regression is a common model for binary classification. Another is inference for Gaussian processes, which are used in many Bayesian models because of their flexibility, and where stochstic gradient Langevin algorithms have been applied [FE15]. A third application is latent Dirichlet allocation (LDA), often used for document classification [BNJ03]. As new documents are published, it is desirable to update the distribution of topics without excessive re-computation.[1]

- **Optimization.** One online optimization method is to sample a point from the exponential of the (weighted) negative loss ([CBL06, HAK07], Lemma 10 in [NR17]). There are settings such as online logistic regression where the only known way to achieve optimal regret is a Bayesian sampling approach [FKL$^+$18], with lower bounds known for the naive convex optimization approach [HKL14].

- **Reinforcement learning (RL).** Thompson sampling [RVRK$^+$18, DFE18] solves RL problems by maximizing the expected reward at each period with respect to a sample from the Bayesian posterior for the environment parameters, reducing it to the online posterior sampling problem.

In all of these applications, because a sample is needed at every epoch $t$, it is desirable to have a fast online sampling algorithm. In particular, the ultimate goal is to design an algorithm for Problem 1.1 such that the number of gradient evaluations is almost *constant* at each epoch $t$, so that the computational requirements at each epoch do not increase over time. This is challenging because at epoch $t$, one has to incorporate information from *all* $t + 1$ functions $f_0, \ldots, f_t$ in roughly $O(1)$ time.

Our main contribution is an algorithm for Problem 1.1 that computes $\widetilde{O}_T(1)$ gradients per epoch, under mild assumptions on the functions[2]. All previous rigorous results (even with comparable assumptions) imply a bound on the number of gradient or function evaluations which is at least linear in $T$; see Table 1. Our assumptions are motivated by real-world considerations and hold in the setting of online Bayesian logistic regression when the data vectors satisfy natural regularity properties.

In the offline setting, our result also implies the first algorithm to sample from a $d$-dimensional log-concave distribution $\propto e^{-\sum_{t=1}^{T} f_t}$ where the $f_t$'s are not assumed strongly convex and the total number of gradient evaluations is roughly $T \log(T) + \text{poly}(d)$, instead of $T \times \text{poly}(d)$ implied by prior works (Table 1).

A natural approach to online sampling is to design a Markov chain with the right steady state distribution [NR17, DMM19, DCWY18, CFM$^+$18]. The main difficulty is that running a step of a Markov chain that incorporates all previous functions takes time $\Omega(t)$ at epoch $t$; all previous algorithms with provable guarantees suffer from this. To overcome this, one must use stochasticity – for example, sample a subset of the previous functions. However, this fails because of the large variance of the gradient. Our result relies on a stochastic gradient Langevin dynamics (SGLD) Markov chain with a carefully designed variance reduction step and fixed batch size.

We emphasize that we do not assume that the functions $f_t$ are strongly convex. This is important for applications such as logistic regression. Even if the negative log-prior $f_0$ is strongly convex, we cannot obtain the same bounds by using existing results on strongly convex $f$, because the bounds depend on the condition number of $\sum_{t=0}^{T} f_t$, which grows as $T$. Lack of strong convexity is a technical barrier to analyzing our Markov chain and, here, our main contribution is a martingale exit time argument that shows that our Markov chain is constrained to a ball of radius roughly $1/\sqrt{t}$ for time that is sufficient for it to reach within $\varepsilon$ of $\pi_t$.

## 2 Our algorithm and results

### 2.1 Assumptions

Denote by $\mathcal{L}(Y)$ the distribution of a random variable $Y$. For any two probability measures $\mu, \nu$, denote the 2-Wasserstein distance by $W_2(\mu, \nu) := \inf_{(X,Y)\sim\Pi(\mu,\nu)} \sqrt{\mathbb{E}[\|X - Y\|^2]}$, where $\Pi(\mu, \nu)$ denotes the set of all possible couplings of random vectors $(\hat{X}, \hat{Y})$ with marginals $\hat{X} \sim \mu$ and $\hat{Y} \sim \nu$. For every $t \in \{0, \ldots, T\}$, define $F_t := \sum_{k=0}^{t} f_k$, and let $x_t^\star$ be a minimizer of $F_t(x)$ on $\mathbb{R}^d$. For any $x \in \mathbb{R}^d$, let $\delta_x$ be the Dirac delta distribution centered at $x$. We make the following assumptions:

**Assumption 1 (Smoothness/Lipschitz gradient (with constants $L_0, L > 0$)).** *For all $1 \leq t \leq T$ and $x, y \in \mathbb{R}^d$, $\|\nabla f_t(y) - \nabla f_t(x)\| \leq L \|x - y\|$. For $t = 0$, $\|\nabla f_0(y) - \nabla f_0(x)\| \leq L_0 \|x - y\|$.*

We allow $f_0$ to satisfy our assumptions with a different parameter value, since in Bayesian applications $f_0$ models a "prior" which has different scaling from $f_1, f_2, \ldots f_T$.

**Assumption 2 (Bounded second moment with exponential concentration (with constants $A, k > 0$, $c \geq 0$)).** *For all $0 \leq t \leq T$ and all $s \geq 0$, $\mathbb{P}_{X\sim\pi_t}(\|X - x_t^\star\| \geq s/\sqrt{t+c}) \leq Ae^{-ks}$.*

Note Assumption 2 implies a bound on the second moment, $m_2^{1/2} := (\mathbb{E}_{x\sim\pi_t} \|x - x_t^\star\|_2^2)^{\frac{1}{2}} \leq C/\sqrt{t+c}$ for $C := (2 + 1/k) \log(A/k^2)$. For conciseness, we write bounds in terms of this parameter $C$.[3]

**Assumption 3 (Drift of mode (with constants $\mathfrak{D} \geq 0$, $c \geq 0$)).** *For all $0 \leq t, \tau \leq T$ such that $\tau \in [t, \max\{2t, 1\}]$, $\|x_t^\star - x_\tau^\star\| \leq \mathfrak{D}/\sqrt{t+c}$.*

Assumption 2 says that the "data is informative enough" – the current distribution $\pi_t$ (posterior) concentrates near the mode $x_t^\star$ as $t$ increases. The $\frac{1}{t}$ decrease in the second moment is what one would expect based on central limit theorems such as the Bernstein-von Mises theorem. Assumption 2 is a weaker condition than strong convexity: if the $f_t$'s are $\alpha$-strongly convex, then $\pi_t(x) \propto e^{-\sum_{k=0}^{t} f_k(x)}$ concentrates to within $\sqrt{d}/\sqrt{\alpha(t+1)}$; however, many distributions satisfy Assumption 2 without being strongly log-concave. For instance, posterior distributions used in Bayesian logistic regression satisfy Assumption 2 under natural conditions on the data, but are not strongly log-concave with comparable parameters (Section 2.4). Hence, together Assumptions 1 and 2 are a weaker condition than strong convexity and gradient Lipschitzness, the typical assumptions under which the offline algorithm is analyzed. Similar to the typical assumptions, our assumptions avoid the "ill-conditioned" case when the distribution becomes more concentrated in one direction than another as the number of functions $t$ increases.

Assumption 3 is typically satisfied in the setting where the $f_t$'s are iid. This is the case when we observe iid random variables and define functions $f_t$ based on them, as will be the case for our application to Bayesian logistic regression (Problem 2.2). To help with intuition, note that Assumption 3 is satisfied for the problem of Gaussian mean estimation: the mode is the same as the mean, and the assumption reduces to the fact that a random walk drifts on the order of $\sqrt{t}$, and hence the mean of the posterior drifts by $O_T(1/\sqrt{t})$, after $t$ time steps. We need this assumption because our algorithm uses cached gradients computed $\Theta_T(t)$ time steps ago, and in order for the past gradients to be close in value to the gradient at the current point, the points where the gradients were last calculated should be at distance $O_T(1/\sqrt{t})$ from the current point. We give a simple example where the assumptions hold (Appendix G of the supplement).

In Section 2.4 we show these assumptions hold for functions arising in online Bayesian logistic regression; unlike previous work on related techniques [NDH+17, CFM+18], our assumptions are weak enough to hold in such applications, as they do not require $f_0, \ldots, f_T$ to be strongly convex.

### 2.2 Algorithm for online sampling

At every epoch $t = 1, \ldots, T$, given gradient access to the functions $f_0, \ldots, f_t$, Algorithm 2 generates a point $X^t$ approximately distributed according to $\pi_t \propto e^{-\sum_{k=0}^{t} f_k(x)}$. It does so by running SAGA-LD (Algorithm 1), with step size $\eta_t$ that decreases as the epoch, and a given number of steps $i_{\max}$.

Our main Theorem 2.1 says that for each sample to have fixed TV error $\varepsilon$, at each epoch the number of steps $i_{\max}$ only needs to be poly-logarithmic in $T$.

Algorithm 1 makes the following update rule at each step for the SGLD Markov chain $X_i$, for a certain choice of stochastic gradient $g_i$, where $\mathbb{E}[g_i] = \sum_{k=0}^{t} \nabla f_k(X_i)$:

$$X_{i+1} = X_i - \eta_t g_i + \sqrt{2\eta_t}\xi_i, \qquad \xi_i \sim N(0, I_d). \tag{1}$$

Key to our algorithm is the construction of the variance reduced stochastic gradient $g_i$. It is constructed by taking the sum of the cached gradients at previous points in the chain and correcting it with a batch of constant size $b$.

This variance reduction is only effective when the points where the cached gradients were computed stay within $\widetilde{O}_T(1/\sqrt{t})$ of the current mode $x_t^{\star}$. Algorithm 2 ensures that this holds with high probability by resetting to the sample at the previous power of 2 if the sample has drifted too far.

The step size $\eta_t$ is determined by an input parameter $\eta_0 > 0$. We set $\eta_t = \eta_0/t+c$ for the following reason: Assumption 2 says that the variance of the target distribution $\pi_t$ decreases at the rate $C^2/t+c$, and we want to ensure that the variance of each step of Langevin dynamics decreases at roughly the same rate. With the step size $\eta_t = \eta_0/t+c$, the Markov chain can travel across a sub-level set containing most of the probability measure of $\pi_t$ in roughly the same number $i_{\max} = \widetilde{O}_T(1)$ of steps at each epoch $t$. We will take the acceptance radius to be $C' = 2.5(C_1 + \mathfrak{D})$ where $C_1$ is given by (65) in the supplement, and show that with good probability this choice of $C'$ ensures $\|X^{t-1} - X^{t'}\| \leq 4(C_1+\mathfrak{D})/\sqrt{t+c}$ in Algorithm 2. Note that in practice, one need not know the values of the regularity constants in Assumptions 1-3 but can instead use heuristics to "tune" the Markov chain's parameters.

---

**Algorithm 1** SAGA-LD

---

**Input:** Oracles for $\nabla f_k$ for $k \in [0, t]$, step size $\eta > 0$, batch size $b \in \mathbb{N}$, number of steps $i_{\max}$, initial point $X_0$, cached gradients $G^k = \nabla f_k(u_k)$ for some points $u_k$, and $s = \sum_{k=1}^{t} G^k$. **Output:** $X_{i_{\max}}$

  1: **for** $i$ from 0 to $i_{\max} - 1$ **do**
  2:    (Sample batch) Sample with replacement a (multi)set $S$ of size $b$ from $\{1, \ldots, t\}$.
  3:    (Calculate gradients) For each $k \in S$, let $G_{\text{new}}^k = \nabla f_k(X_i)$.
  4:    (Variance-reduced gradient estimate) Let $g_i = \nabla f_0(X_i) + s + \frac{t}{b}\sum_{k \in S}(G_{\text{new}}^k - G^k)$.
  5:    (Langevin step) Let $X_{i+1} = X_i - \eta g_i + \sqrt{2\eta}\xi_i$ where $\xi_i \sim N(0, I)$.
  6:    (Update sum) Update $s \leftarrow s + \sum_{k \in \text{set}(S)}(G_{\text{new}}^k - G^k)$.
  7:    (Update gradients) For each $k \in S$, update $G^k \leftarrow G_{\text{new}}^k$.
  8: **end for**

---

### 2.3 Result in the online setting

In this section we give our main result for the online sampling problem; for additional results in the offline sampling problem, see Appendix A in the supplement.

**Theorem 2.1** (Online variance-reduced SGLD). *Suppose that $f_0, \ldots, f_T : \mathbb{R}^d \to \mathbb{R}$ are (weakly) convex and satisfy Assumptions 1-3 with $c = L_0/L$. Let $C = (2 + 1/k)\log(A/k^2)$. Then there exist parameters $b = 9$, $\eta_0 = \widetilde{\Theta}\left(\frac{\varepsilon^4}{L^2 \log^6(T)(C+\mathfrak{D})^2 d}\right)$, and $i_{\max} = \widetilde{O}\left(\frac{(C+\mathfrak{D})^2 \log^2(T)}{\eta_0 \varepsilon^2}\right)$, such that at each epoch $t$, Algorithm 2 generates an $\varepsilon$-approximate independent sample $\mathsf{X}^t$ from $\pi_t$.[4] The total number of gradient evaluations $i_{\max}$ required at each epoch $t$ is polynomial in $d, L, C, \mathfrak{D}, \varepsilon^{-1}$ and $\log(T)$. Here, $\widetilde{\Theta}$ and $\widetilde{O}$ hide polylogarithmic factors in $d, L, C, \mathfrak{D}, \varepsilon^{-1}$ and $\log(T)$.*

Note that the dependence of $i_{\max}$ on $\varepsilon$ is $i_{\max} = \widetilde{O}_{\varepsilon}\left(\frac{1}{\varepsilon^6}\right)$. See Section B.4 in the supplement for the proof of Theorem 2.1. Note that the algorithm needs to know the parameters, but bounds are enough.

Previous results all imply a bound on the number of gradient or function evaluations[5] at each epoch which is at least linear in $T$. Our result is the first to obtain bounds on the number of gradient

**Algorithm 2** Online SAGA-LD

---

**Input:** $T \in \mathbb{N}$ and gradient oracles for functions $f_t : \mathbb{R}^d \to \mathbb{R}$, for all $t \in \{0, \ldots, T\}$ , where only the gradient oracles $\nabla f_0, \ldots, \nabla f_t$ are available at epoch $t$, an initial point $\mathsf{X}^0 \in \mathbb{R}^d$.
**Input:** step size $\eta_0 > 0$, batch size $b > 0$, $i_{\max} > 0$, constant offset $c$, acceptance radius $C'$.
**Output:** At each epoch $t$, a sample $\mathsf{X}^t$

1: Set $s = 0$.                                                             ▷ Initial gradient sum
2: **for** epoch $t = 1$ to $T$ **do**
3:      Set $t' = 2^{\lfloor \log_2(t-1) \rfloor}$ if $t > 1$, and $t' = 0$ if $t = 1$.        ▷ The previous power of 2
4:      **if** $\left\| \mathsf{X}^{t-1} - \mathsf{X}^{t'} \right\| \leq C'/\sqrt{t+c}$ **then** $\mathsf{X}_0^t \hookleftarrow \mathsf{X}^{t-1}$ ▷ If the previous sample hasn't drifted too far, use the previous sample as warm start
5:      **else** $\mathsf{X}_0^t \hookleftarrow \mathsf{X}^{t'}$        ▷ If the previous sample has drifted too far, reset to the sample at time $t'$
6:      **end if**
7:      Set $G_t \hookleftarrow \nabla f_t(\mathsf{X}_0^t)$
8:      Set $s \hookleftarrow s + G_t$.
9:      For all gradients $G_k = \nabla f_k(u_k)$ which were last updated at time $t/2$, replace them by $\nabla f_k(\mathsf{X}_0^t)$ and update $s$ accordingly.
10:      Draw $i_t$ uniformly from $\{1, \ldots, i_{\max}\}$.
11:      Run Algorithm 1 with step size $\eta_0/t+c$, batch size $b$, number of steps $i_t$, initial point $\mathsf{X}_0^t$, and precomputed gradients $G_k$ with sum $s$. Keep track of when the gradients are updated.
12:      Return the output $\mathsf{X}^t = \mathsf{X}_{i_t}^t$ of Algorithm 1.
13: **end for**

---

evaluations which are poly-logarithmic, rather than linear, in $T$ at each epoch. We are able to do better by exploiting the sum structure of $-\sum_{k=0}^{t} f_t$ and the fact that the $\pi_t$ evolve slowly. See Section 4 for a detailed comparison.

## 2.4 Application to Bayesian logistic regression

Next, we show that Assumptions 1-3, and therefore Theorem 2.1, hold in the setting of online Bayesian logistic regression, when the data satisfy certain regularity properties. Logistic regression is a fundamental and widely used model in Bayesian statistics [AC93]. It has served as a model problem for methods in scalable Bayesian inference [WT11, HCB16, CB19, CB18], of which online sampling is one approach. Additionally, sampling from the logistic regression posterior is the key step in the optimal algorithm for online logistic regret minimization [FKL+18].

In Bayesian logistic regression, one models the data $(u_t \in \mathbb{R}^d, y_t \in \{-1, 1\})$ as follows: there is some unknown $\theta_0 \in \mathbb{R}^d$ such that given $u_t$ (the "independent variable"), for all $t \in \{1, \ldots, T\}$ the "dependent variable" $y_t$ follows a Bernoulli distribution with "success" probability $\phi(u_t^\top \theta)$ ($y_t = 1$ with probability $\phi(u_t^\top \theta)$ and $-1$ otherwise) where $\phi(x) := 1/(1+e^{-x})$. The problem we consider is:

**Problem 2.2** (**Bayesian logistic regression**). *Suppose the $y_t$'s are generated from $u_t$'s as Bernoulli random variables with "success" probability $\phi(u_t^\top \theta)$. At every epoch $t \in \{1, \ldots, T\}$, after observing $(u_k, y_k)_{k=1}^{t}$, return a sample from the posterior distribution[6] $\hat{\pi}_t(\theta) \propto e^{-\sum_{k=0}^{t} \hat{f}_k(\theta)}$, where $\hat{f}_0(\theta) := e^{-\alpha \|\theta\|^2/2}$ and $\hat{f}_k(\theta) := -\log[\phi(y_k u_k^\top \theta)]$.*

We show that Algorithm 2 succeeds for Bayesian logistic regression under reasonable conditions on the data-generating distribution – namely, that inputs are bounded and we see data in all directions.[7]

**Theorem 2.3** (**Online Bayesian logistic regression**). *Suppose that for some $\mathfrak{B}, M, \sigma > 0$, we have $\|\theta_0\| \leq \mathfrak{B}$ and that $u_t \sim P_u$ are iid, where $P_u$ is a distribution satisfying the following: For $u \sim P_u$, (1) $\|u\| \leq M$ ("bounded") and (2) $\mathbb{E}_u[uu^\top \mathbb{1}_{|u^\top \theta_0| \leq 2}] \succeq \sigma I_d$ ("restricted" covariance matrix is bounded away from 0). Then for the functions $\hat{f}_0, \ldots, \hat{f}_T$ in Problem 2.2, and any $\varepsilon > 0$, there exist parameters $L, \log(A), k^{-1}, \mathfrak{D} = \mathrm{poly}(M, \sigma^{-1}, \alpha, \mathfrak{B}, d, \varepsilon^{-1}, \log(T))$ such that Assumptions 1, 2, and 3 hold for all $t$ with probability at least $1 - \varepsilon$. Therefore Alg. 2 gives $\varepsilon$-approximate samples from $\pi_t$ for $t \in [1, T]$ with $\mathrm{poly}(M, \sigma^{-1}, \alpha, \mathfrak{B}, d, \varepsilon^{-1}, \log(T))$ gradient evaluations at each epoch.*

In Section 5 we show that in numerical simulations, our algorithm achieves competitive accuracy with the same runtime compared to an algorithm specialized to logistic regression, the Pólya-Gamma sampler. However, the Pólya-Gamma sampler has two drawbacks: its running time at each epoch scales linearly as $t$ (our algorithm scales as $\text{polylog}(t)$), and it is unknown whether Pólya-Gamma attains TV-error $\varepsilon$ in time polynomial in $\frac{1}{\varepsilon}$, $t$, $d$, and other problem parameters.

# 3 Proof overview for online problem

For the online problem, information theoretic constraints require us to use "information" from at least $\Omega(t)$ gradients to sample with fixed TV error at the $t$'th epoch (see Appendix H). Thus, to use only $\widetilde{O}_T(1)$ gradients at each epoch, we must reuse gradient information from past epochs. We accomplish this by reusing gradients computed at points in the Markov chain, including points at past epochs. This saves a factor of $T$ over naive SGLD, but only if we can show that these past points in the chain track the distributions' mode, and that our chain stays close to the mode (Lemma B.2 in supplement).

The distribution is concentrated to $O_T(1/\sqrt{t})$ at the $t$th epoch (Assumption 2), and we need the Markov chain to stay within $\widetilde{O}_T(1/\sqrt{t})$ of the mode. The bulk of the proof (Lemma B.3 in supplement) is to show that with high probability (w.h.p.) the chain stays within this ball. Once we establish that the Markov chain stays close, we combine our bounds with existing results on SGLD from [DMM19] to show that we only need $\widetilde{O}_T(1)$ steps per epoch (Lemma B.6). Finally, an induction with careful choice of constants finishes the proof (Theorem 2.1). Details of each of these steps follow.

**Bounding the variance of the stochastic gradient (see Lemma B.2).** We reduce the variance of our stochastic gradient by using the gradient evaluated at a past point $u_k$ and estimating the difference in the gradients between our current point $X_i^t$ and past point $u_k$. Using the $L$-Lipschitz property (Assumption 1) of the gradients, we show that the variance of this stochastic gradient is bounded by $\frac{t^2 L^2}{b} \max_k \|X_i^t - u_k\|^2$. To obtain this bound, observe that the individual components $\{\nabla f_k(X_i^t) - \nabla f_k(u_k)\}_{k \in S}$ of the stochastic gradient $g_i^t$ have variance at most $= t^2 L^2 \max_k \|X_i^t - u_k\|^2$ by the Lipschitz property. Averaging with a batch saves a factor of $b$. For the number of gradient evaluations to stay nearly constant at each step, increasing the batch size is not a viable option to decrease our stochastic gradient's variance. Rather, showing that $\|X_i^t - u_k\|$ decreases as $\|X_i^t - u_k\| = \widetilde{O}_T(1/\sqrt{t})$, implies the variance of our stochastic gradient decreases at each epoch at the desired rate.

**Bounding the escape time from a ball where the stochastic gradient has low variance (see Lemma B.3).** Our main challenge is to bound the distance $\|X_i - u_k\|$. Because we do not assume strong convexity, we cannot use proof techniques of past papers analyzing variance-reduced SGLD methods. [CFM+18, NDH+17] used strong convexity to show that w.h.p., the Markov chain does not travel too far from its initial point, implying a bound on the variance of their stochastic gradients. Unfortunately, many important applications, including logistic regression, lack strong convexity.

To deal with the lack of strong convexity, we instead use a martingale exit time argument to show that the Markov chain remains inside a ball of radius $r = \widetilde{O}_T(1/\sqrt{t})$ w.h.p. for a large enough time $i_{\max}$ for the Markov chain to reach a point within TV distance $\varepsilon$ of the target distribution. Towards this end, we would like to bound the distance from the current state of the Markov chain to the mode $\|X_i^t - x_t^\star\|$ by $\widetilde{O}_T(1/\sqrt{t})$, and bound $\|x_t^\star - u_k\|$ by $\widetilde{O}_T(1/\sqrt{t})$. Together, this allows us to bound the distance $\|X_i^t - u_k\| = O_T(1/\sqrt{t})$. We can then use our bound on $\|X_i^t - u_k\| = \widetilde{O}_T(1/\sqrt{t})$ together with Lemma B.2 to bound the variance of the stochastic gradient by roughly $\widetilde{O}_T(1/t)$.

*Bounding $\|x_t^\star - u_k\|$.* Since $u_k$ is a point of the Markov chain, possibly at a previous epoch $\tau \leq t$, roughly speaking we can bound this distance inductively by using bounds obtained at the previous epoch $\tau$ (Lemma B.6). Noting that $u_k = X_i^\tau$ for some $i \leq i_{\max}$, we use our bound for $\|u_k - x_\tau^\star\| = O_T(1/\sqrt{\tau}) = O_T(1/\sqrt{t})$ obtained at the previous epoch $\tau$, together with Assumption 3 which says that $\|x_t^\star - x_\tau^\star\| = O_T(1/\sqrt{t})$, to bound $\|x_t^\star - u_k\|$.

*Bounding $\|X_i^t - x_t^\star\|$.* To bound the distance $\rho_i := \|X_i^t - x_t^\star\|$ to the mode, we would like to bound the increase $\rho_{i+1} - \rho_i$ at each step $i$ in the Markov chain. Unfortunately, the expected increase in the distance $\|X_i^t - x_t^\star\|$ is much larger when the Markov chain is close to the mode than when it is far away from the mode, making it difficult to get a tight bound on the increase in the distance at each step. To get around this problem, we instead use a martingale exit time argument on $\|X_i^t - x_t^\star\|^2$, the

*squared* distance from the current state of the Markov chain to the mode. The advantage in using squared distance is that the expected increase in squared distance due to the Gaussian noise term $\sqrt{2\eta_t}\xi_i$ in the Markov chain update rule (Equation (1)) is the same regardless of the position of the chain, allowing us to obtain tighter bounds on the increase regardless of the Markov chain's current position. We then use weak convexity to bound the component of the increase in $\left\|X_i^t - x_\tau^\star\right\|^2$ that is due to the gradient term $-\eta_t g_i$, and apply Azuma's martingale concentration inequality to bound the exit time from the ball, showing the chain remains at distance of roughly $\widetilde{O}_T(1/\sqrt{t})$ from the mode.

**Bounding the TV error (Lemma B.6).** We now show that if $u_k$ is close to $x_\tau^\star$, then $\mathsf{X}^t$ will be a good sample from $\pi_t$. More precisely, we show that if at epoch $t$ the Markov chain starts at $X_0^t$ such that $\left\|X_0^t - x_\tau^\star\right\| \leq \Re/\sqrt{t+c}$ ($\Re$ to be chosen later), then $\left\|\mathcal{L}(X_{i_{\max}}^t) - \pi_t\right\|_{\mathrm{TV}} \leq O(\varepsilon/\log_2(T))$.

To do this, we use two bounds: a bound on the Wasserstein distance between the initial point $X_0^t$ and the target density $\pi_t$, and a bound on the variance of the stochastic gradient. We then plug the bounds into Corollary 18 of [DMM19] (reproduced as Theorem B.4 in the supplementary material), to show that $i_{\max} = \widetilde{O}_{\varepsilon,T}(\mathrm{poly}(1/\varepsilon))$ steps per epoch are sufficient to obtain a bound of $\varepsilon$ on the TV error.

**Bounding the number of gradient evaluations at each epoch.** Working out constants, we see $i_{\max} = \mathrm{poly}(d, L, C, \mathfrak{D}, \varepsilon^{-1}, \log(T))$ suffices to obtain TV-error $\varepsilon$ each epoch. A constant batch size suffices, so the total number of evaluations is $O(i_{\max}b) = \mathrm{poly}(d, L, C, \mathfrak{D}, \varepsilon^{-1}, \log(T))$.

# 4   Related work

**Online convex optimization.**   Our motivation for studying the online sampling problem comes partly from the successes of online (convex) optimization [Haz16]. In online convex optimization, one chooses a point $x_t \in K$ at each step and suffers a loss $f_t(x_t)$, where $K$ is a compact convex set and $f_t : K \to \mathbb{R}$ is a convex function [Zin03]. The aim is to minimize the regret compared to the best point in hindsight, where $\mathrm{Regret}_T = \sum_{t=1}^T f_t(x_t) - \min_{x^*} \sum_{t=1}^T f_t(x^*)$. The same offline convex optimization algorithms such as gradient descent and Newton's method can be adapted to the online setting [Zin03, HAK07].

**Online sampling.** To the best of our knowledge, all previous algorithms with provable guarantees in our setting require computation time that grows polynomially with $t$. This is because any Markov chain taking all previous data into account needs $\Omega_T(t)$ gradient (or function) evaluations per step. On the other hand, there are many streaming algorithms that are used in practice which lack provable guarantees, or which rely on properties of the data (such as compressibility [HCB16, CB19]).

The most relevant theoretical work in our direction is [NR17]. The authors consider a changing log-concave distribution on a convex body, and show that under certain conditions, they can use the previous sample as a warm start and only take a constant number of steps of their Dikin walk chain at each stage. They consider the online sampling problem in the more general setting where the distribution is restricted to a convex body. However, [NR17] do not achieve optimal results in our setting, since they do not separately consider the case when $F_t = \sum_{k=0}^t f_k$ has a sum structure and therefore require $\Omega(t)$ function evaluations at epoch $t$. Moreover, they do not consider how concentration properties of the distribution translate into more efficient sampling. When the $f_t$ are linear, they need $O_T(1)$ steps and $O_T(t)$ evaluations per epoch. However, in the general convex setting with smooth $f_t$'s, they need $O_T(t)$ steps per epoch and $O_T(t^2)$ evaluations per epoch.

There are many other online sampling and other approaches to estimating changing distributions, used in practice. The *Laplace approximation*, perhaps the simplest, approximates the posterior distribution with a Gaussian [BDT16]; however, most distributions cannot be well-approximated by Gaussians. *Stochastic gradient Langevin dynamics* [WT11] can be used in an online setting; however, it suffers from large variance which we address in this work. The *particle filter* [DMHW+12, GDM+17] is a general algorithm to track changing distributions. Another approach (besides sampling) is *variational inference*, which has also been considered in an online setting ([WPB11], [BBW+13]).

**Variance reduction techniques.** Variance reduction techniques for SGLD were initially proposed in [DRW+16], when sampling from a fixed distribution $\pi \propto e^{-\sum_{t=0}^T f_t}$. [DRW+16] propose two variance-reduced SGLD techniques, CV-ULD and SAGA-LD. CV-ULD re-computes the full gradient $\nabla F$ at an "anchor" point every $r$ steps and updates the gradient at intermediate steps by subsampling the difference in the gradients between the current point and the anchor point. SAGA-LD, on the

| Algorithm | oracle calls per epoch | Other assumptions |
|---|---|---|
| Online Dikin walk [NR17, §5.1] | $O_T(T)$ | **Strong convexity** Bounded ratio of densities |
| Langevin [DMM19, DCWY18] | $O_T(T)$ | — |
| SGLD [DMM19] | $O_T(T)$ | — |
| SAGA-LD [CFM+18] | $O_T(T)$ | **Strong convexity** Lipschitz Hessian |
| CV-ULD [CFM+18] | $O_T(T)$ | **Strong convexity** |
| **This work** | $\mathrm{polylog}(T)$ | bounded second moment bounded drift of minimizer |

Table 1: Bounds on the number of gradient (or function) evaluations required by different algorithms to solve the online sampling problem. Lipschitz gradient is assumed for all algorithms. [NR17] analyzed the online Dikin walk for a different setting where the target has compact support; here we give the result one should obtain for support $\mathbb{R}^d$, where it reduces to the ball walk. Thus it is possible the assumptions we give for the online Dikin walk can be weakened. Note that the number of gradient or function evaluations for the basic Langevin and SGLD algorithms and online Dikin walk depend multiplicatively on $T$ (i.e., $T \times \mathrm{poly}(d, L, \text{other parameters})$), while the number of evaluations for variance-reduced SGLD methods depend only additively on $T$ (i.e., $T + \mathrm{poly}(d, L, \text{other parameters})$).

other hand, keeps track of when each gradient $\nabla f_t$ was computed, and updates individual gradients with respect to when they were last computed. [CFM+18] show that CV-ULD can sample in the offline setting in roughly $T + d^2/\varepsilon (L/m)^6$ gradient evaluations, and that SAGA-LD can sample in $T + T^{\sqrt{d}}/\varepsilon (L/m)^{3/2}(1 + L_H)$ evaluations, where $L_H$ is the Lipschitz constant of the Hessian of $-\log(\pi)$.[8]

## 5 Simulations

We test our algorithm against other sampling algorithms on a synthetic dataset for logistic regression. The dataset consists of $T = 1000$ data points in dimension $d = 20$. We compare the marginal accuracies of the algorithms.

The data is generated as follows. First, $\theta \sim N(0, I_d), b \sim N(0, 1)$ are randomly generated. For each $1 \le t \le T$, a feature vector $x_t \in \mathbb{R}^d$ and output $y_t \in \{0, 1\}$ are generated by

$$x_{t,i} \sim \text{Bernoulli}\left(\frac{s}{d}\right) \qquad 1 \le i \le d, \tag{2}$$

$$y_t \sim \text{Bernoulli}(\phi(\theta^\top x_t + b)), \tag{3}$$

where the sparsity is $s = 5$ in our simulations, and $\phi(x) = \frac{1}{1+e^{-x}}$ is the logistic function. We chose $x_t \in \{0, 1\}^d$ because in applications, features are often indicators.

The algorithms are tested in an online setting as follows. At epoch $t$ each algorithm has access to $x_{s,i}, y_s$ for $s \le t$, and attempts to generate a sample from the posterior distribution $p_t(\theta) \propto e^{-\frac{\|\theta\|^2}{2}} e^{-\frac{b^2}{2}} \prod_{s=1}^t \phi(\theta^\top x_t + b)$; the time is limited to $t = 0.1$ seconds. We estimate the quality of the samples at $t = T = 1000$, by saving the state of the algorithm at $t = T - 1$, and re-running it 1000 times to collect 1000 samples. We replicate this entire simulation 8 times, and the marginal accuracies of the runs are given in Figure 1.

The marginal accuracy (MA) is a heuristic to compare accuracy of samplers (see e.g. [DMS17], [FOW11] and [CR+17]). The marginal accuracy between the measure $\mu$ of a sample and the target $\pi$ is $MA(\mu, \pi) := 1 - \frac{1}{2d} \sum_{i=1}^d \|\mu_i - \pi_i\|_{\mathrm{TV}}$, where $\mu_i$ and $\pi_i$ are the marginal distributions of $\mu$ and $\pi$ for the coordinate $x_i$. Since MALA is known to sample from the correct stationary distribution for the class of distributions analyzed in this paper, we let $\pi$ be the estimate of the true distribution obtained from 1000 samples generated from running MALA for a long time (1000 steps). We estimate the TV distance by the TV distance between the histograms when the bin widths are 0.25 times the sample standard deviation for the corresponding coordinate of $\pi$.

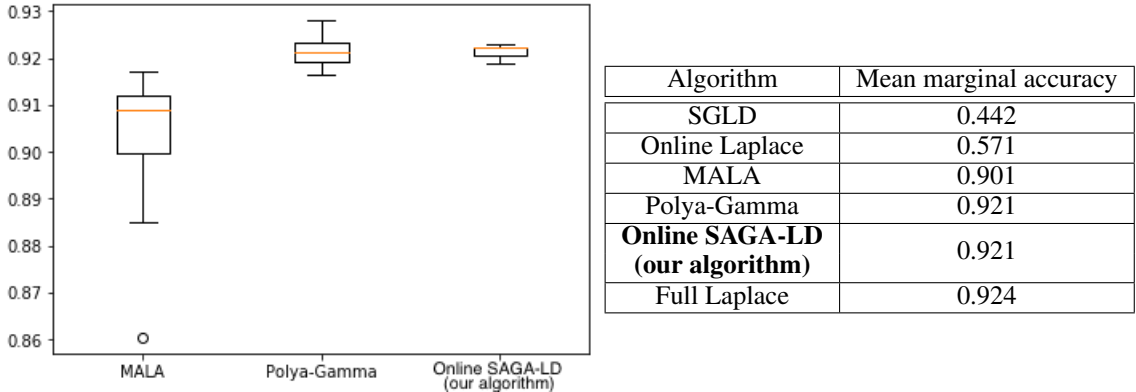

| Algorithm | Mean marginal accuracy |
|---|---|
| SGLD | 0.442 |
| Online Laplace | 0.571 |
| MALA | 0.901 |
| Polya-Gamma | 0.921 |
| **Online SAGA-LD (our algorithm)** | 0.921 |
| Full Laplace | 0.924 |

Figure 1: Marginal accuracies of 5 different sampling algorithms on online logistic regression, with $T = 1000$ data points, dimension $d = 20$, and time 0.1 seconds, averaged over 8 runs. SGLD and online Laplace perform much worse and are not pictured.

We compare our online SAGA-LD algorithm with SGLD, full and online Laplace approximation, Pólya-Gamma, and MALA. The Laplace method approximates the target distribution with a multivariate Gaussian distribution. Here, one first finds the mode of the target distribution using a deterministic optimization technique and then computes the Hessian $\nabla^2 F_t$ of the log-posterior at the mode. The inverse of this Hessian is the covariance matrix of the Gaussian. In the online version of the algorithm, given in [CL11], to speed up optimization, only a quadratic approximation (with diagonal Hessian) to the log-posterior is maintained. The Pólya-Gamma chain [DFE18] is a Markov chain specialized to sample from the posterior for logistic regression. Note that in contrast, our algorithm works more generally for any smooth probability distribution over $\mathbb{R}^d$.

Our results show that our online SAGA-LD algorithm is competitive with the best samplers for logistic regression, namely, the Pólya-Gamma Markov chain and the full Laplace approximation. We note that the full Laplace approximation requires optimizing a sum of $t$ functions, which has runtime that scales linearly with $t$ at each epoch, while our method only scales as $\mathrm{polylog}(t)$.

The parameters are as follows. The step size at epoch $t$ is $\frac{0.1}{1+0.5t}$ for MALA, $\frac{0.01}{1+0.5t}$ for SGLD, and $\frac{0.05}{1+0.5t}$ for online SAGA-LD. A smaller step size must be used with SGLD because of the increased variance. For MALA, a larger step size can be used because the Metropolis-Hastings acceptance step ensures the stationary distribution is correct. The batch size for SGLD and online SAGA-LD is 64. The step sizes $\eta_0$ were chosen by hand from testing various values in the range from 0.001 to 1.0. We found the reset step of our online SAGA-LD algorithm, and the random number of steps, to be unnecessary in practice, so the results are reported for our online SAGA-LD algorithm without these features. The experiments were run on Fujitsu CX2570 M2 servers with dual, 14-core 2.4GHz Intel Xeon E5 2680 v4 processors with 384GB RAM running the Springdale distribution of Linux.

## 6    Conclusion and future work

In this paper we obtain logarithmic-in-$T$ bounds at each epoch when sampling from a sequence of log-concave distributions $\pi_t \propto e^{-\sum_{k=0}^{t} f_k}$, improving on previous results which are linear-in-$T$ in the online setting. Since we do not assume the $f_t$'s are strongly convex, we also obtain bounds which have an improved dependence on $T$ for a wider range of applications including Bayesian logistic regression. While our assumption of Lipschitz gradients requires the target to have full support on $\mathbb{R}^d$, one can also consider extending our $\mathrm{polylog}(T)$ bounds to log-densities supported on a compact set. Restricting the distribution to have compact support can cause the target distribution's covariance matrix to become increasingly ill-conditioned as the number of functions $t$ increases. To overcome this, we could modify our algorithm by including an adaptive pre-conditioner which changes along with the target distribution.

### Acknowledgments

This research was partially supported by NSF CCF-1908347 and SNSF 200021_182527 grants.

## Footnotes

[1]Note that LDA requires sampling from non-logconcave distributions. Our algorithm can be used for non-logconcave distributions, but our theoretical guarantees are only for logconcave distributions.

[2]The subscript $T$ in $\widetilde{O}_T$ means that we only show the dependence on the parameters $t, T$, and exclude dependence on non-$T$, $t$ parameters such as the dimension $d$, sampling accuracy $\varepsilon$ and the regularity parameters $C, \mathfrak{D}, L$ which we define in Section 2.1.

[3]Having a bounded second moment suffices to obtain (weaker) polynomial bounds (by replacing the use of the concentration inequality with Chebyshev's inequality). We use this slightly stronger condition because exponential concentration improves the dependence on $\varepsilon$, and is typically satisfied in practice.

[4]See Definition B.1 in the supplement for the formal definition. Necessarily, $\|\mathcal{L}(\mathsf{X}^t) - \pi_t\|_{\text{TV}} \leq \varepsilon$.

[5]In our setting a gradient can be computed in at worst $2d$ function evaluations. In many applications (including logistic regression) gradient evaluation takes the same number of operations as function evaluation.

[6]Here we use a Gaussian prior but this can be replaced by any $e^{-f_0}$ where $f_0$ is strongly convex and smooth.

[7]For simplicity, we state the result (Theorem 2.3) in the case where the input variables $u$ are iid, but note that the result holds more generally (see Lemma E.1 in the supplement for a more general statement of our result).

[8]The bounds of [CFM+18] are given for sampling within a specified Wasserstein error, not TV error. The bounds we give here are the number of gradient evaluations one would need if one samples with Wasserstein error $\widetilde{\varepsilon}$ which roughly corresponds to TV error $\varepsilon$; roughly, one requires $\widetilde{\varepsilon} = O(\varepsilon/\sqrt{T})$ to sample with TV error $\varepsilon$.

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
