[Supplementary Material · Online_Sampling_NeurIPS_CameraReady-13-34.pdf]

# Supplementary Appendix for "Online Sampling from Log-Concave Distributions"

## A    Results in the offline setting

In the offline setting, we have access to all the $f_t$'s from the start. Our goal is simply to generate a sample from the single target distribution $\pi_T(x) \propto e^{-\sum_{t=1}^{T} f_t(x)}$ with TV error $\varepsilon$. Since we do not assume that the $f_t$'s are given in any particular order, we replace Assumption 2 which depends on the order in which the functions are given, with an assumption (Assumption 4) on the target $\sum_{t=1}^{T} f_t(x)$ which does not depend on the $f_t$'s ordering. In place of working with the sequence of distributions $\pi_1, \pi_2 \ldots$ which depend on the $f_t$'s ordering, we introduce an inverse temperature parameter $\beta > 0$ and consider the distributions $\pi_T^\beta(x) \propto e^{-\beta \sum_{t=1}^{T} f_t(x)}$. In place of Assumption 2, we assume:

**Assumption 4 (Bounded second moment with exponential concentration (with constants $A, k > 0$)).** *For all $\frac{1}{T} \leq \beta \leq 1$ and all $s \geq 0$, $\mathbb{P}_{X \sim \pi_T^\beta}(\|X - x^\star\| \geq \frac{s}{\sqrt{\beta T}}) \leq A e^{-ks}$.*

Assumption 4 says the distributions $\pi_T^\beta$ become more concentrated as $\beta$ increases from $1/T$ to 1. By sampling from a sequence of distributions $\pi_T^\beta$ where we gradually increase $\beta$ from $1/T$ to 1 at each epoch, our offline algorithm (Algorithm 3 in the supplementary material) is able to approach the target distribution $\pi_T = \pi_T^1$ when starting from a cold start that is far from a sublevel set containing most of the probability measure of $\pi_T$, without requiring strong convexity. Moreover, since scaling by $\beta$ does not change the location of the minimizer $x^\star$ of $\beta \sum_{t=1}^{T} f_t(x)$, we can drop Assumption 3.

**Theorem A.1 (Offline variance-reduced SGLD).** *Suppose that $f_1, \ldots, f_T$ satisfy Assumptions 1 and 4. Then there exist $b$, $\eta$, and $i_{\max}$ which are polynomial in $d, L, C, \varepsilon^{-1}$ and poly-logarithmic in $T$, such that Algorithm 3 generates a sample $X^T$ such that $\|\mathcal{L}(X^T) - \pi_T\|_{\mathrm{TV}} \leq \varepsilon$. Moreover, the total number of gradient evaluations is $\mathrm{polylog}(T) \times \mathrm{poly}(d, L, C, \mathfrak{D}, \varepsilon^{-1}) + \widetilde{O}(T)$.*

See Theorem D.2 for precise dependencies. The theorem could also be stated with a $f_0$, but we omitted it for simplicity. As in the online setting, we do not assume strong convexity. Further, our additive dependence on $T$ in Theorem A.1 is tight up to log factors, since the number of gradient evaluations needed to sample from a distribution satisfying Assumptions 1-3 is at least $\Omega(T)$ due to information theoretic requirements (we show this informally in supplementary Appendix H).

Compared to previous work in this setting, our results are the first to obtain an additive dependence on $T$ and polynomial dependence on the other parameters without assuming strong convexity. While the results of [CFM$^+$18] for SAGA-LD and CV-LD have additive dependence on $T$, their results require the functions $f_1, \ldots, f_T$ to be strongly convex. Since the basic Dikin walk and basic Langevin algorithms compute all $T$ functions or all $T$ gradients every time the Markov chain takes a step, and the number of steps in their Markov chain depends polynomially on the other parameters such as $d$ and $L$, the number of gradient (or function) evaluations required by these algorithms is *multiplicative* in $T$. Even though the basic SGLD algorithm computes a mini-batch of the gradients at each step, roughly speaking the batch size at *each step* of the chain should be $\Omega_T(T)$ for the stochastic gradient to have the required variance, implying that basic SGLD also has multiplicative dependence on $T$.

## B    Proof of online theorem (Theorem 2.1)

First we formally define what we mean by "almost independent".

**Definition B.1.** *We say that $X^1, \ldots, X^T$ are $\varepsilon$-**approximate independent samples** from probability distributions $\pi_1, \ldots, \pi_T$ if for independent random variables $Y_t \sim \pi_t$, there exists a coupling between $(X^1, \ldots, X^T)$ and $(Y^1, \ldots, Y^T)$ such that for each $t \in [1, T]$, $X^t = Y^t$ with probability $1 - \varepsilon$.*

### B.1    Bounding the variance of the stochastic gradient

We first show that the variance reduction in Algorithm 2 reduces the variance from the order of $t^2$ to $t^2 \|x - x'\|^2$, where $x'$ is a past point. This will be on the order of $t$ if we can ensure

$\|x - x'\| = O_T\left(\frac{1}{\sqrt{t}}\right)$. Later, we will bound the probability of the bad event that $\|x - x'\|$ becomes too large.

**Lemma B.2.** *Fix $x \in \mathbb{R}^d$ and $\{u_k\}_{1 \le k \le t}$ and let $S$ be a multiset chosen with replacement from $\{1, \ldots, t\}$. Let*

$$g^t = \nabla f_0(x) + \left[\sum_{k=1}^{t} \nabla f_k(u_k)\right] + \frac{t}{b}\sum_{k \in S}[\nabla f_k(x) - \nabla f_k(u_k)]. \tag{4}$$

*Then*

$$\mathbb{E}\left[\left\|g^t - \sum_{k=0}^{t} \nabla f_k(x)\right\|^2\right] \le \frac{t^2}{b}L^2 \max_k \|x - u_k\|^2 \tag{5}$$

$$\left\|g^t - \sum_{k=0}^{t} \nabla f_k(x)\right\|^2 \le 4t^2 L^2 \max_k \|x - u_k\|^2. \tag{6}$$

*Proof.* Let $V$ be the random variable given by

$$V = \frac{t}{b}\left[(\nabla f_k(u_k) - \nabla f_k(x)) - \underset{k \in [t]}{\mathbb{E}}[\nabla f_k(u_k) - \nabla f_k(x)]\right], \tag{7}$$

where $k \in [t]$ is chosen uniformly at random. Let $V_1, \ldots, V_b$ be independent draws of $V$. Note that the distribution of $\left\|g^t - \sum_{k=0}^{t} \nabla f_k(x)\right\|^2$ is the same as that of $\left\|\sum_{j=1}^{b} V_j\right\|^2$. Because the $V_j$ are independent,

$$\mathbb{E}\left[\left\|g^t - \sum_{k=0}^{t} \nabla f_k(x)\right\|^2\right] = \mathbb{E}\left[\left\|\sum_{j=1}^{b} V_j\right\|^2\right] = \mathrm{tr}\left(\mathbb{E}\left[\left(\sum_{j=1}^{b} V_j\right)\left(\sum_{j=1}^{b} V_j\right)^\top\right]\right) \tag{8}$$

$$= \mathrm{tr}\left(\mathbb{E}\left[\sum_{j=1}^{b} V_j V_j^\top\right]\right) = \sum_{j=1}^{b} \mathbb{E}\left[\mathrm{tr}(V_j V_j^\top)\right] = b\mathbb{E}[\|V\|^2]. \tag{9}$$

We calculate

$$\mathbb{E}[\|V\|^2] = \frac{t^2}{b^2}\mathrm{Var}_{k \in [t]}\left(\nabla f_k(u_k) - \nabla f_k(x)\right) \tag{10}$$

$$\le \frac{t^2}{b^2}\left(\underset{k \in [t]}{\mathbb{E}}\left[\|\nabla f_k(u_k) - \nabla f_k(x)\|^2\right]\right) \tag{11}$$

$$\le \frac{t^2}{b^2}L^2 \max_k \|x - u_k\|^2. \tag{12}$$

Combining (9) and (12) gives the first part.

The final part follows because (12) implies $\left\|\sum_{j=1}^{b} V_j\right\|^2 \le 4t^2 L^2 \max_k \|x - u_k\|^2$.  $\square$

## B.2  Bounding the escape time from a ball

**Lemma B.3.** *Suppose that the following hold:*

1. *$F : \mathbb{R}^d \to \mathbb{R}$ is convex, differentiable, and $L$-smooth, with a minimizer $x^\star \in \mathbb{R}^d$.*

2. *$\zeta_i$ is a random variable depending only on $X_0, \ldots, X_i$ such that $\mathbb{E}[\zeta_i | X_0, \ldots, X_i] = 0$, and whenever $\|X_j - x^\star\| \le r$ for all $j \le i$, $\|\zeta_i\| \le S$.*

*Let $X_0$ be such that $\|X_0 - x^\star\| \le r$ and define $X_i$ recursively by*

$$X_{i+1} = X_i - \eta g_i + \sqrt{\eta_t}\xi_i \tag{13}$$
$$\text{where } g_i = \nabla F(X_i) + \zeta_i \tag{14}$$
$$\xi_i \sim N(0, I_d), \tag{15}$$

*and define the event* $G := \{\|X_j - x^\star\| \le r \ \forall 1 \le j \le i_{\max}\}$. *Then for* $r^2 > \|X_0 - x^\star\|^2 + i_{\max}[2\eta^2(S^2 + L^2r^2) + \eta d]$ *and* $C_\xi \ge \sqrt{2d}$,

$$\mathbb{P}(G^c) \le i_{\max}\left[\exp\left(-\frac{(r^2 - \|X_0 - x^\star\|^2 - i_{\max}[2\eta^2(S^2 + L^2r^2) + \eta d])^2}{2i_{\max}(2\eta Sr + 2\sqrt{\eta}C_\xi(r + \eta S + \eta Lr) + \eta C_\xi^2)^2}\right)\right. \tag{16}$$

$$\left.+ \exp\left(-\frac{C_\xi^2 - d}{8}\right)\right]. \tag{17}$$

*Proof.* Note that if $\|x - x^\star\| \le r$, then because $F$ is $L$-smooth, $\|\nabla F(x)\| \le L\|x - x^\star\| \le Lr$. If $\|X_i - x^\star\| \le r$ and $\|\zeta_i\| \le S$, then

$$\|X_{i+1} - x^\star\|^2 - \|X_i - x^\star\|^2 \tag{18}$$

$$= \|X_i - x^\star - \eta g_i + \sqrt{\eta}\xi_i\|^2 - \|X_i - x^\star\|^2 \tag{19}$$

$$= -2\eta\langle g_i, X_i - x^\star\rangle + \eta^2\|g_i\|^2 + 2\sqrt{\eta}\langle X_i - x^\star - \eta g_i, \xi_i\rangle + \eta\|\xi_i\|^2 \tag{20}$$

$$= \underbrace{-2\eta\langle \nabla F_t(X_i), X_i - x^\star\rangle}_{\le 0 \text{ by convexity}} -2\eta\langle \zeta_i, X_i - x^\star\rangle + \eta^2\|g_i\|^2 + 2\sqrt{\eta}\langle X_i - x^\star - \eta g_i, \xi_i\rangle + \eta\|\xi_i\|^2 \tag{21}$$

$$\le -2\eta\langle \zeta_i, X_i - x^\star\rangle + 2\eta^2\left(\|\nabla F(x_i)\|^2 + \|\zeta_i\|^2\right) + 2\sqrt{\eta}\langle X_i - x^\star - \eta g_i, \xi_i\rangle + \eta\|\xi_i\|^2 \tag{22}$$

$$\le -2\eta\langle \zeta_i, X_i - x^\star\rangle + 2\eta^2(L^2r^2 + S^2) + 2\sqrt{\eta}\langle X_i - x^\star - \eta g_i, \xi_i\rangle + \eta\|\xi_i\|^2 \tag{23}$$

$$= 2\eta^2(L^2r^2 + S^2) + \eta d \underbrace{-2\eta\langle \zeta_i, X_i - x^\star\rangle + 2\sqrt{\eta}\langle X_i - x^\star - \eta g_i, \xi_i\rangle + \eta(\|\xi_i\|^2 - d)}_{(*)}. \tag{24}$$

Note that $(*)$ has expectation 0 conditioned on $X_0, \dots, X_i$. To use Azuma's inequality, we need our random variables to be bounded. Also, recall that we assumed $\|X_i - x^\star\|$ is bounded above by $r$. Thus, we define a toy Markov chain coupled to $X_i$ as follows. Let $X_0' = X_0$ and

$$X_{i+1}' = \begin{cases} X_i', & \text{if } \|X_i' - x^\star\| \ge r \\ X_i' - \eta g_i + \sqrt{\eta}\xi_i', & \text{otherwise} \end{cases} \tag{25}$$

$$\text{where } g_i = \nabla F(X_i') + \zeta_i, \tag{26}$$

$$\xi_i' = \min(C_\xi, \|\xi_i\|)\frac{\xi_i}{\|\xi_i\|}, \tag{27}$$

$$\xi_i \sim N(0, I_d). \tag{28}$$

Then $Y_i' := \|X_i' - x^\star\|^2 - i[2\eta^2(S^2 + L^2r^2) + \eta d]$ is a supermartingale with differences upper-bounded by

$$Y_{i+1}' - Y_i' \le \begin{cases} 0, & \|X_i' - x^\star\| \ge r \\ -2\eta\langle \zeta_i, X_i' - x^\star\rangle + 2\sqrt{\eta}\langle X_i' - x^\star - \eta g_i, \xi_i'\rangle + \eta(\|\xi_i\|^2 - d), & \|X_i' - x^\star\| < r \end{cases} \tag{29}$$

$$\le 2\eta Sr + 2\sqrt{\eta}(r + \eta(S + Lr))C_\xi + \eta(C_\xi^2 - d) \tag{30}$$

$$\le 2\eta Sr + 2\sqrt{\eta}C_\xi(r + \eta S + \eta Lr) + \eta C_\xi^2. \tag{31}$$

By Azuma's inequality, for $\lambda > 0$ and for $r^2 > \|X_0 - x^\star\|^2 + i[2\eta^2(S^2 + L^2r^2) + \eta d]$,

$$\mathbb{P}\left(\|X_i' - x^\star\|^2 - \|X_0 - x^\star\|^2 - i[2\eta^2(S^2 + L^2r^2) + \eta d] > \lambda\right) \tag{32}$$

$$\le \exp\left(-\frac{\lambda^2}{2i(2\eta Sr + 2\sqrt{\eta}C_\xi(r + \eta S + \eta Lr) + \eta C_\xi^2)^2}\right) \tag{33}$$

$$\implies \mathbb{P}\left(\|X_i' - x^\star\| > r\right) \tag{34}$$

$$\le \exp\left(-\frac{(r^2 - \|X_0 - x^\star\|^2 - i[2\eta^2(S^2 + L^2r^2) + \eta d])^2}{2i(2\eta Sr + 2\sqrt{\eta}C_\xi(r + \eta S + \eta Lr) + \eta C_\xi^2)^2}\right). \tag{35}$$

If $\|X_i - x^\star\| \geq r$ for some $i \leq i_{\max}$, then either $\|X_i' - x^\star\| \geq r$ for some $i \leq i_{\max}$, or $X_i$ otherwise becomes different from $X_i'$, which happens only when $\xi_i \geq C_\xi$ for some $i \leq i_{\max}$. Thus by the Hanson-Wright inequality, since $C_\xi \geq \sqrt{2d}$,

$$\mathbb{P}\left(\mathcal{I} \leq i_{\max}\right) \tag{36}$$

$$\leq \sum_{i=1}^{i_{\max}} \mathbb{P}(\|X_i' - x^\star\|^2 > r^2) + \sum_{i=1}^{i_{\max}} \mathbb{P}(\|\xi_i\| > C_\xi) \tag{37}$$

$$\leq i_{\max} \left[ \exp\left( -\frac{(r^2 - \|X_0 - x^\star\|^2 - i_{\max}[2\eta^2(S^2 + L^2 r^2) + \eta d])^2}{2 i_{\max}(2\eta S r + 2\sqrt{\eta} C_\xi(r + \eta S + \eta L r) + \eta C_\xi^2)^2} \right) \right. \tag{38}$$

$$\left. + \exp\left( -\frac{C_\xi^2 - d}{8} \right) \right]. \tag{39}$$

$\square$

## B.3 Bounding the TV error

Lemma B.6 will allow us to carry out the induction step for the proof of the main theorem.

We will use the following result of [DMM19]. Note that this result works more generally with non-smooth functions, but we will only consider smooth functions. Their algorithm, Stochastic Proximal Gradient Langevin Dynamics, reduces to SGLD in the smooth case. We will apply this Lemma with our variance-reduced stochastic gradients in Algorithm 1.

**Lemma B.4** ([DMM19], Corollary 18). *Suppose that $f : \mathbb{R}^d \to \mathbb{R}$ is convex and $L$-smooth. Let $\mathcal{F}_i$ be a filtration with $\xi_i$ and $g(x_i)$ defined on $\mathcal{F}_i$, and satisfying $\mathbb{E}[g(x_i)|\mathcal{F}_{i-1}] = \nabla f(x_i)$, $\sup_x \mathrm{Var}[g(x)|\mathcal{F}_{i-1}] \leq \sigma^2 < \infty$. Consider SGLD for $f(x)$ run with step size $\eta$ and stochastic gradient $g(x)$, with initial distribution $\mu_0$ and step size $\eta$; that is,*

$$x_{i+1} = x_i - \eta g(x_i) + \sqrt{\eta} \xi_i, \qquad\qquad \xi_i \sim N(0, I). \tag{40}$$

*Let $\mu_n$ denote the distribution of $x_n$ and let $\pi$ be the distribution such that $\pi \propto e^{-f}$. Suppose*

$$\eta \leq \min\left\{ \frac{\varepsilon}{2(Ld + \sigma^2)}, \frac{1}{L} \right\}, \tag{41}$$

$$n \geq \left\lceil \frac{W_2^2(\mu_0, \pi)}{\eta \varepsilon} \right\rceil. \tag{42}$$

*Let $\overline{\mu} = \frac{1}{n} \sum_{k=1}^n \mu_k$ be the "averaged" distribution. Then $\mathrm{KL}(\overline{\mu}|\pi) \leq \varepsilon$.*

**Remark B.5.** *The result in [DMM19] is stated when $g(x)$ is independent of the history $\mathcal{F}_i$, but the proof works when the stochastic gradient is allowed to depend on history, as in SAGA. For SAGA, $\mathcal{F}_i$ contains all the information up to time step $i$, including which gradients were replaced at each time step.*

*Note [DMM19] is derived by analogy to online convex optimization. The optimization guarantees are only given at the point $\overline{x}$ equal to the average of the $x_t$ (by Jensen's inequality). For the sampling problem, this corresponds to selecting a point from the averaged distribution $\overline{\mu}$.*

Define the good events

$$G_t = \left\{ \forall s \leq t, \forall 0 \leq i \leq i_s, \|X_i^s - x_s^\star\| \leq \frac{\mathfrak{R}}{\sqrt{s + L_0/L}} \right\}, \tag{43}$$

$$H_t = \left\{ \forall s \leq t \text{ s.t. } s \text{ is a power of 2 or } s = 0, \|X^s - x_s^\star\| \leq \frac{C_1}{\sqrt{s + L_0/L}} \right\}. \tag{44}$$

$G_t$ is the event that the Markov chain never drifts too far from the current mode (which we want, in order to bound the stochastic gradient of SAGA), and $H_t$ is the event that the samples at powers of 2 are close to the respective modes (which we want because we will use them as reset points). Roughly, $G_t^c$ will involve union-bounding over bad events whose probabilities we will set to be $O\left(\frac{\varepsilon}{T}\right)$ and $H_t^c$ will involve union-bounding over bad events whose probabilities we will set to be $O\left(\frac{\varepsilon}{\log_2(T)}\right)$.

**Lemma B.6** (Induction step). *Suppose that Assumptions 1, 2, and 3 hold with $c = \frac{L_0}{L}$ and $L_0 \geq L$. Let $X_i^\tau$ be obtained by running Algorithm 2 with $C' = 2.5(C_1 + \mathfrak{D})$, $C_1 \geq C$, and $\mathfrak{R} \geq 2(C_1 + \mathfrak{D})$. Suppose $\eta_t = \frac{\eta_0}{t + L_0/L}$ and $\varepsilon_2 > 0$ is such that*

$$\eta_0 \leq \frac{\varepsilon_2^2}{Ld + 9L^2(\mathfrak{R} + \mathfrak{D})^2/b}, \qquad i_{\max} \geq \frac{20(C_1 + \mathfrak{D})^2}{\eta_0 \varepsilon_2^2}. \tag{45}$$

*Suppose $\varepsilon_1 > 0$ is such that for any $\tau \geq 1$,*

$$\mathbb{P}\left(G_\tau | G_{\tau-1} \cap H_{\tau-1}\right) \geq 1 - \varepsilon_1. \tag{46}$$

*Suppose $t$ is a power of 2. Then the following hold.*

1. *For $t < \tau \leq 2t$, $\mathbb{P}(G_\tau | G_t \cap H_t) \geq 1 - (\tau - t)\varepsilon_1$.*

2. *Fix $X_i^s$ for $s \leq t, 0 \leq i \leq i_{\max}$ such that $G_t \cap H_t$ holds (i.e., condition on the filtration $\mathcal{F}_t$ on which the algorithm is defined). Then*

$$\|\mathcal{L}(X^\tau) - \pi_\tau\|_{TV} \leq (\tau - t)\varepsilon_1 + \varepsilon_2. \tag{47}$$

3. *We have for $\tau = 2t$,*

$$\mathbb{P}\left(G_\tau \cap H_\tau | G_t \cap H_t\right) \geq 1 - (t\varepsilon_1 + \varepsilon_2 + Ae^{-kC_1}). \tag{48}$$

*These also hold in the case $t = 0$ and $\tau = 1$, when $L_0 \geq L$.*

*Proof.* Let $F_t(x) = \sum_{k=0}^t f_k(x)$.

First, note that $H_{\tau-1} = \cdots = H_t$, because $H_s$ is defined as an intersection of events with indices $\leq s$, that are powers of 2. (See (44).) Moreover, $G_\tau$ is a subset of $G_{\tau-1}$ for each $\tau$, by (43).

**Proof of Statement 1.** The first statement holds by induction on $\tau$ and assumption on $\varepsilon_1$. We need to show $P(G_\tau^c | G_t \cap H_t) \leq (\tau - t)\varepsilon_1$ by induction. Assuming it is true for $\tau$, we have by the union bound that

$$\mathbb{P}(G_{\tau+1}^c | G_t, H_t) \leq \mathbb{P}(G_{\tau+1}^c \cap G_\tau | G_t \cap H_t) + \mathbb{P}(G_\tau^c | G_t \cap H_t) \tag{49}$$

$$\leq \mathbb{P}(G_{\tau+1}^c | G_\tau \cap G_t \cap H_t) + \mathbb{P}(G_\tau^c | G_t \cap H_t). \tag{50}$$

Now the event $G_\tau \cap G_t \cap H_t$ is the same as the event $G_\tau \cap H_\tau$, by the previous paragraph. Thus this is $\leq \varepsilon + (\tau - t)\varepsilon$, completing the induction step.

**Proof of Statement 2.** For the second statement, note that for $t < \tau \leq 2t$,

$$\|X_0^\tau - x_\tau^\star\| \leq \|X_0^\tau - X^t\| + \|X^t - x_t^\star\| + \|X_t^\star - x_\tau^\star\| \tag{51}$$

$$\leq \frac{2.5(C_1 + \mathfrak{D})}{\sqrt{\tau + L_0/L}} + \frac{C_1}{\sqrt{t + L_0/L}} + \frac{\mathfrak{D}}{\sqrt{t + L_0/L}} \tag{52}$$

$$\leq \frac{4(C_1 + \mathfrak{D})}{\sqrt{\tau + L_0/L}}. \tag{53}$$

where in the 2nd inequality we used that

1. Algorithm 2 ensures that $\|X_0^\tau - X^t\| \leq \frac{C'}{\sqrt{\tau + L_0/L}} = \frac{2.5(C_1 + \mathfrak{D})}{\sqrt{\tau + L_0/L}}$ (The algorithm resets $X_0^\tau$ to $X^t$ if $\|X_0^\tau - X^t\|$ is greater than $\frac{C'}{\sqrt{\tau + L_0/L}}$, making the term 0. This is the place where the resetting is used.),

2. the definition of $H_t$, and

3. the drift assumption (Assumption 3).

In the 3rd inequality we used that $\sqrt{t} \ge \sqrt{\tau/2} \ge \sqrt{\tau}/1.5$.

Therefore

$$W_2^2(\delta_{X_0^\tau}, \pi_\tau) \le 2\left\|X_0^\tau - x_\tau^\star\right\|^2 + 2W_2^2(\delta_{x_\tau}, \pi_\tau) \le \frac{32(C_1 + \mathfrak{D})^2}{\tau + L_0/L} + \frac{2C^2}{\tau + L_0/L} \le \frac{40(C_1 + \mathfrak{D})^2}{\tau + L_0/L}. \tag{54}$$

where the second moment bound comes from Assumption 2 and $C \le C_1$.

Define a toy Markov chain coupled to $X_i^\tau$ as follows. Let $\widetilde{X}_j^s = X_j^s$ for $s < \tau$, $\widetilde{X}_0^\tau = X_0^\tau$, and

$$\widetilde{X}_{i+1}^\tau = \begin{cases} \widetilde{X}_i^\tau - \eta g_i^\tau + \sqrt{\eta}\xi_i, & \text{when } \left\|\widetilde{X}_j^\tau - x_\tau^\star\right\| \le \frac{\mathfrak{R}}{\sqrt{\tau + L_0/L}} \text{ for all } 0 \le j \le i \\ \widetilde{X}_i^\tau - \eta\nabla F_\tau(\widetilde{X}_i), & \text{otherwise.} \end{cases} \tag{55}$$

where $g_i^\tau$ is the stochastic gradient for $\widetilde{X}_i^\tau$ in Algorithm 1 and $\xi_i \sim N(0, I_d)$. By Lemma B.2, the variance of $g_i^\tau$ is at most $\frac{\tau^2 L^2}{b} \max_{(\frac{\tau+1}{2}, 0) \le (s,j) \le (\tau,i)} \left\|\widetilde{X}_i^\tau - \widetilde{X}_j^s\right\|^2$. (The ordering on ordered pairs is lexicographic. Note $s > \frac{t}{2}$ because Algorithm 2 refreshes all gradients that were updated at time $\frac{t}{2}$.) If the first case of (55) always holds, we bound (using the condition that $G_t$ holds)

$$\left\|\widetilde{X}_i^\tau - \widetilde{X}_j^s\right\| \le \left\|\widetilde{X}_i^\tau - x_\tau^\star\right\| + \left\|x_\tau^\star - x_s^\star\right\| + \left\|x_s^\star - \widetilde{X}_j^s\right\| \tag{56}$$

$$\le \frac{\mathfrak{R}}{\sqrt{\tau + L_0/L}} + \frac{\mathfrak{D}}{\sqrt{s + L_0/L}} + \frac{\mathfrak{R}}{\sqrt{s + L_0/L}} \tag{57}$$

$$\le \frac{3\mathfrak{R} + 2\mathfrak{D}}{\sqrt{\tau + L_0/L}} < \frac{3(\mathfrak{R} + \mathfrak{D})}{\sqrt{\tau + L_0/L}} \tag{58}$$

$$\implies \frac{\tau^2 L^2}{b} \max_{(\frac{t+1}{2}, 0) \le (s,j) \le (\tau,i)} \left\|\widetilde{X}_i^\tau - \widetilde{X}_j^s\right\|^2 \le \frac{9\tau L^2(\mathfrak{R} + \mathfrak{D})^2}{b}. \tag{59}$$

We can apply Lemma B.4 with $\varepsilon = 2\varepsilon_2^2$, $L \hookleftarrow L(\tau + L_0/L)$, $\sigma^2 \le \frac{9\tau L^2(\mathfrak{R}+\mathfrak{D})^2}{b}$, $W_2^2(\mu_0, \pi) \le \frac{40(C_1 + \mathfrak{D})^2}{\tau + L_0/L}$. Note that $\eta_\tau \le \frac{\varepsilon_2^2}{(\tau + L_0/L)(Ld + 9L^2(\mathfrak{R}+\mathfrak{D})^2/b)} \le \frac{\varepsilon_2^2}{(\tau L + L_0)d + 9L^2\tau(\mathfrak{R}+\mathfrak{D})^2/b}$ does satisfy (41), as $F_\tau = \sum_{k=0}^\tau f_k$ is $(\tau L + L_0)$-smooth by Assumption 1. Let $i \in [i_{\max}]$ be uniform random on $[i_{\max}]$, and $\widetilde{X}^\tau = \widetilde{X}_i^\tau$; note that the distribution $\widetilde{\mu}$ of $\widetilde{X}^\tau$ is the mixture distribution of $\widetilde{X}_1^\tau, \ldots, \widetilde{X}_{i_{\max}}^\tau$. Under the conditions on $\eta, i_{\max}$, by Pinsker's inequality and Lemma B.4,

$$\|\mathcal{L}(\widetilde{X}^\tau) - \pi_\tau\|_{\mathrm{TV}} \le \sqrt{\frac{1}{2}\mathrm{KL}(\widetilde{\mu}|\pi_\tau)} \le \varepsilon_2. \tag{60}$$

Note that under $G_\tau$, $X_i^s = \widetilde{X}_i^s$ for all $i \le i_{\max}$ and $s \le \tau$, so

$$\|\mathcal{L}(X^\tau) - \pi_\tau\|_{\mathrm{TV}} \le \mathbb{P}(G_\tau^c | \mathcal{F}_t) + \|\mathcal{L}(\widetilde{X}_i^\tau) - \pi_\tau\|_{\mathrm{TV}} \le (\tau - t)\varepsilon_1 + \varepsilon_2. \tag{61}$$

This shows Statement 2.

**Proof of Statement 3.** For Statement 3, note that by Assumption 2,

$$\mathbb{P}_{X \sim \pi_{2t}}\left[\|X - x_{2t}^\star\| \ge \frac{C_1}{\sqrt{2t + L_0/L}}\right] \le Ae^{-kC_1}. \tag{62}$$

Combining (61) and (62) for $\tau = 2t$ gives (48).

Finally, note that the proof goes through when $t = 0$, $\tau = 1$. $\qquad\square$

## B.4 Setting the constants; Proof of main theorem

*Proof of Theorem 2.1.* We set the parameters $\eta_0, i_{\max}$ of Algorithm 2, as follows:

$$\varepsilon_1 = \frac{\varepsilon}{3T}, \tag{63}$$

$$\varepsilon_2 = \frac{\varepsilon}{3\lceil \log_2(T) + 1 \rceil}, \tag{64}$$

$$C_1 = \left(2 + \frac{1}{k}\right) \log\left(\frac{A}{\varepsilon_2 k^2}\right), \tag{65}$$

$$\mathfrak{R} = \frac{10000(C_1 + \mathfrak{D})\sqrt{d}}{\varepsilon_2} \log\left(\max\left\{L, C_1 + \mathfrak{D}, \frac{1}{\varepsilon_1}\right\}\right), \tag{66}$$

$$\eta_0 = \frac{\varepsilon_2^2}{2L^2(\mathfrak{R} + \mathfrak{D})^2}, \tag{67}$$

$$i_{\max} = \left\lceil \frac{20(C_1 + \mathfrak{D})^2}{\eta_0 \varepsilon_2^2} \right\rceil = \left\lceil \frac{40L^2(\mathfrak{R} + \mathfrak{D})^2(C_1 + \mathfrak{D})^2}{\varepsilon_2^4} \right\rceil. \tag{68}$$

We can check that $\eta_0 = \widetilde{\Theta}\left(\frac{\varepsilon^4}{L^2 \log^6(T)(C+\mathfrak{D})^2 d}\right)$, and $i_{\max} = \widetilde{O}\left(\frac{(C+\mathfrak{D})^2 \log^2(T)}{\eta_0 \varepsilon^2}\right)$ (where $\widetilde{\Theta}$ and $\widetilde{O}$ hide polylogarithmic dependence on $d, L, C, \mathfrak{D}, \varepsilon^{-1}$ and $\log(T)$, as claimed in Theorem 2.1. The constants have not been optimized.

We will choose parameters and prove by induction that for $t = 2^a$, $a \in \mathbb{N}_0$, $t \leq T$,

$$\mathbb{P}(G_t \cap H_t) \geq 1 - t\varepsilon_1 - 2(a+1)\varepsilon_2. \tag{69}$$

We will also show that (69) implies that if $t = 2^a + b$ for $0 < b \leq 2^a$,

$$\mathbb{P}(G_t \cap H_{2^a}) \geq 1 - t\varepsilon_1 - 2(a+1)\varepsilon_2, \tag{70}$$

$$\|\mathcal{L}(X_t) - \pi_t\|_{\text{TV}} \leq t\varepsilon_1 + (2a+3)\varepsilon_2. \tag{71}$$

With the values of $\varepsilon_1$ and $\varepsilon_2$, (71) gives the theorem, except for the $\varepsilon$-approximate independence of the samples. To obtain approximate independence, note that the distribution of $X^t$ conditioned on the filtration $\mathcal{F}_1 \subseteq \cdots \subseteq \mathcal{F}_{t-1}$, where the filtration $\mathcal{F}_\tau$ includes both the random batch $S$ as well as the points in the Markov chain up to time $\tau$, satisfies $\|(\mathcal{L}(X^t)|\mathcal{F}_{t-1}) - \pi_t\|_{\text{TV}} \leq t\varepsilon_1 + (2a+3)\varepsilon_2$. This implies that the samples $X^1, X^2, \ldots, X^t$ are $\varepsilon$-approximately independent with $\varepsilon = t\varepsilon_1 + (2a+3)\varepsilon_2$.

Let $\eta_0, \mathfrak{R}$ be constants to be chosen, and for any $t \in \mathbb{N}$, let

$$\eta_t = \frac{\eta_0}{t + L_0/L}, \tag{72}$$

$$r_t = \frac{\mathfrak{R}}{\sqrt{t + L_0/L}}, \tag{73}$$

$$S_t = 6\sqrt{t}L(\mathfrak{R} + \mathfrak{D}), \tag{74}$$

We claim that it suffices to choose parameters so that the following hold for each $t$, $1 \leq t \leq T$, and some $C_\xi \geq \sqrt{2d}$:

$$\varepsilon_1 \geq i_{\max} \left[ \exp\left(-\frac{\left(r_t^2 - \frac{16(C_1+\mathfrak{D})^2}{t+L_0/L} - i_{\max}[2\eta_t^2(S_t^2 + L^2 t^2 r_t^2) + \eta_t d]\right)^2}{2i_{\max}(2\eta_t S_t r_t + 2\sqrt{\eta_t}C_\xi(r_t + \eta_t S_t + \eta_t L(t + L_0/L)r_t) + \eta_t C_\xi^2)^2}\right) \right. \tag{75}$$

$$\left. + \exp\left(-\frac{C_\xi^2 - d}{8}\right) \right], \tag{76}$$

$$\eta_0 \leq \frac{\varepsilon_2^2}{Ld + 9L^2(\mathfrak{R} + \mathfrak{D})^2/b}, \tag{77}$$

$$i_{\max} \geq \frac{20(C_1 + \mathfrak{D})^2}{\eta_0 \varepsilon_2^2}, \tag{78}$$

$$Ae^{-kC_1} \le \varepsilon_2, \tag{79}$$

$$C_1 \ge \left(2 + \frac{1}{k}\right) \log\left(\frac{A}{\varepsilon_2 k^2}\right). \tag{80}$$

We first complete the proof assuming that these inequalities hold. Then we show that with the parameter settings in (63)–(68), these inequalities hold.

Suppose that for some $t < T$ the inequalities (75)-(80) hold and the event $G_t \cap H_t$ occurs. We will apply Lemma B.3 to the call of the SAGA-LD algorithm in Algorithm 2, at epoch $t+1$ with $F(x) = \sum_{s=0}^{t+1} f_s(x)$, to show that the conditions of Lemma B.6 are satisfied with $r_{t+1}$ and $S_{t+1}$. We will then apply Lemma B.6 inductively to complete the proof of Theorem 2.1.

We first show that the assumption (46) of Lemma B.6 is satisfied for any $\varepsilon_1$ satisfying inequality (75). The first condition of Lemma B.3 holds by assumption on the $f_s$'s. To see that the second condition holds for the values $r_{t+1}$ and $S_{t+1}$, note that by (58) and Lemma B.2, when the event $G_t \cap H_t$ occurs, and when $\left\|X_{t+1}^i - x_{t+1}^\star\right\| \le r_{t+1}$, the stochastic gradient $g_i^{t+1}$ in (55) satisfies $\left\|g_i^{t+1}\right\| \le S_{t+1}$. Therefore, by Lemma B.3 and by inequality (75) we have $\mathbb{P}\left(G_{t+1}|G_t \cap H_t\right) \ge 1 - \varepsilon_1$. Hence, we have that inequality (46) of Lemma B.6 is satisfied for any $\varepsilon_1$ satisfying inequality (75).

Next, we note that assumption (45) of Lemma B.6 is satisfied since Inequalities (77), (78), and (80) ensure that $\eta_0$, $i_{\max}$, and $C$ satisfy the inequalities in (45).

Therefore we have that all the conditions of Lemma B.6 are satisfied. Recall we are proving (69) by induction for $t = 2^a$. By the above, we know we can apply Lemma B.6 for any $t < T$.

**Base case of induction.** We show (69) holds for $t = 1$. By assumption $\left\|X^0 - x_0^\star\right\| \le \frac{C_1}{\sqrt{L_0/L}}$ so $H_0$ holds and the $t = 0$ case of Lemma B.6 shows $\mathbb{P}(G_1) \ge 1 - \varepsilon_1$ and $\mathbb{P}(G_1 \cap H_1) \ge 1 - (\varepsilon_1 + \varepsilon_2 + Ae^{-kC_1}) \ge 1 - (\varepsilon_1 + 2\varepsilon_2)$, using (79) for the last inequality.

**(69) implies (70), (71).** This follows from parts 1 and 2 of Lemma B.6, as follows. Let $A_t = G_t \cap H_t$. Let $t = 2^a + b, 0 < b \le 2^a$.

For (70), using part 1 of Lemma B.6 and the induction hypothesis,

$$\mathbb{P}((G_t \cap H_{2^a})^c) \le \mathbb{P}(G_t^c | A_{2^a}) + \mathbb{P}(A_{2^a}^c) \tag{81}$$

$$\le (t - 2^a)\varepsilon_1 + [2^a \varepsilon_1 + 2(a+1)\varepsilon_2] = t\varepsilon_1 + 2(a+1)\varepsilon_2. \tag{82}$$

For (71), note that by part 2 of of Lemma B.6, conditioned on $A_{2^a}$, $\|\mathcal{L}(X_t) - \pi_t\|_{TV} \le (t - 2^a)\varepsilon_1 + \varepsilon_2$. Without the conditioning,

$$\|\mathcal{L}(X_t) - \pi_t\|_{TV} \le [(t - 2^a)\varepsilon_1 + \varepsilon_2] + \mathbb{P}(A_{2^a}^c) \tag{83}$$

$$\le [(t - 2^a)\varepsilon_1 + \varepsilon_2] + [2^a \varepsilon_1 + 2(a+1)\varepsilon_2] \le 2^a \varepsilon_1 + (2a+3)\varepsilon_2. \tag{84}$$

**Induction step.** We show that if (69) holds for $t$, then it holds for $2t$. We work with the complements. By a union bound,

$$\mathbb{P}(A_{2t}^c) \le \mathbb{P}(A_{2t}^c \cap A_t) + \mathbb{P}(A_t^c) \le \mathbb{P}(A_{2t}^c | A_t) + \mathbb{P}(A_t^c). \tag{85}$$

The first term is bounded by Part 3 of Lemma B.6 and (79), $P(A_{2t}^c | A_t) \le t\varepsilon_1 + \varepsilon_2 + \varepsilon_2$. The second term is bounded by the induction hypothesis, which says $P(A_t^c) \le t\varepsilon_1 + 2(a+1)\varepsilon_2$. Combining these gives $P(A_{2t}^c) \le 2t\varepsilon_1 + 2(a+2)\varepsilon_2$, completing the induction step.

**Showing inequalities.** Setting $C_1$, $\eta_0$, and $i_{\max}$ as in (65), (67), and (68) (with $\mathfrak{R}$ to be determined), we get that (77), (78), and (79) are satisfied, as $\mathfrak{R} \ge \sqrt{\frac{d}{L}}$, $b \ge 9$ imply $\frac{\varepsilon_2^2}{2L^2(\mathfrak{R}+\mathfrak{D})^2} \le \frac{\varepsilon_2^2}{Ld+9L^2(\mathfrak{R}+\mathfrak{D})^2/b}$. Moreover, setting $C_\xi = \sqrt{2d + 8\log\left(\frac{2i_{\max}}{\varepsilon_1}\right)}$ makes $i_{\max}\exp\left(-\frac{C_\xi^2 - d}{8}\right) \le \frac{\varepsilon_1}{2}$. It suffices to show that our choice of $\mathfrak{R}$ makes

$$\frac{\varepsilon_1}{2i_{\max}} \ge \exp\left(-\frac{(r^2 - \frac{16(C_1+\mathfrak{D})^2}{t+L_0/L} - i_{\max}[2\eta_t^2(S_t^2 + L^2(t+L_0/L)^2 r_t^2) + \eta_t d])^2}{2i_{\max}(2\eta_t S_t r_t + 2\sqrt{\eta_t}C_\xi(r_t + \eta_t S_t + \eta_t L(t+L_0/L)r_t) + \eta_t C_\xi^2)^2}\right) \tag{86}$$

It suffices to show

$$\log\left(\frac{2i_{\max}}{\varepsilon_1}\right) \le \frac{(r_t^2 - \frac{16(C_1+\mathfrak{D})^2}{t+L_0/L} - i_{\max}[2\eta_t^2(S_t^2 + L^2(t+L_0/L)^2 r_t^2) + \eta_t d])^2}{2i_{\max}(2\eta_t S_t r_t + 2\sqrt{\eta_t} C_\xi(r_t + \eta_t S_t + \eta_t L(t+L_0/L)r_t) + \eta_t C_\xi^2)^2} \tag{87}$$

$$\Leftarrow r_t^2 \ge \sqrt{2i_{\max}}\left(2\eta_t S_t r_t + 2\sqrt{\eta_t}C_\xi(r_t + \eta_t S_t + \eta_t L(t+L_0/L)r_t) + \eta_t C_\xi^2\right)\sqrt{\log\left(\frac{2i_{\max}}{\varepsilon_1}\right)} \tag{88}$$

$$+ \frac{16(C_1+\mathfrak{D})^2}{t+L_0/L} + i_{\max}[2\eta_t^2(S_t^2 + L^2(t+L_0/L)^2 r_t^2) + \eta_t d] \tag{89}$$

Substituting (72), (73), and (74), this is equivalent to

$$\frac{\mathfrak{R}^2}{t+\frac{L_0}{L}} \ge \frac{\sqrt{2i_{\max}}\eta_0}{t+\frac{L_0}{L}}\Bigg[\left(\frac{2\sqrt{\eta_0}6\sqrt{t}L(\mathfrak{R}+\mathfrak{D})\mathfrak{R}}{\sqrt{t+\frac{L_0}{L}}} + 2C_\xi\left(\mathfrak{R} + \frac{\eta_0 6\sqrt{t}L(\mathfrak{R}+\mathfrak{D})}{\sqrt{t+\frac{L_0}{L}}} + \eta_0 L\mathfrak{R}\right)\right. \tag{90}$$

$$\left. + \sqrt{\eta_0}C_\xi^2\right)\sqrt{\log\left(\frac{2i_{\max}}{\varepsilon_1}\right)} \tag{91}$$

$$+ \frac{16(C_1+\mathfrak{D})^2}{t+\frac{L_0}{L}} + \frac{i_{\max}\eta_0}{t+\frac{L_0}{L}}\left[\frac{2\eta_0}{t+\frac{L_0}{L}}\left(36tL^2(\mathfrak{R}+\mathfrak{D})^2 + L^2\left(t+\frac{L_0}{L}\right)\mathfrak{R}^2\right) + d\right]\Bigg] \tag{92}$$

$$\Leftarrow \mathfrak{R}^2 \ge \sqrt{2i_{\max}\eta_0}(12\sqrt{\eta_0}L(\mathfrak{R}+\mathfrak{D})\mathfrak{R} + 2C_\xi(\mathfrak{R} + 6\eta_0 L(\mathfrak{R}+\mathfrak{D}) + \eta_0 L\mathfrak{R}) \tag{93}$$

$$+ \sqrt{\eta_0}C_\xi^2)\sqrt{\log\left(\frac{2i_{\max}}{\varepsilon_1}\right)} \tag{94}$$

$$+ 16(C_1+\mathfrak{D})^2 + i_{\max}\eta_0\left[\frac{2\eta_0}{t+\frac{L_0}{L}}(36tL^2(\mathfrak{R}+\mathfrak{D})^2 + L^2\left(t+\frac{L_0}{L}\right)\mathfrak{R}^2) + d\right] \tag{95}$$

Using $\eta_0 = \frac{\varepsilon_2^2}{2L^2\mathfrak{R}^2}$, $i_{\max} = \left\lfloor\frac{20(C_1+\mathfrak{D})^2}{\eta_0\varepsilon_2^2}\right\rfloor \le \frac{40(C_1+\mathfrak{D})^2}{\eta_0\varepsilon_2^2}$, and $i_{\max}\eta_0 \le \frac{40(C_1+\mathfrak{D})^2}{\varepsilon_2^2}$, the RHS is at most

$$\sqrt{2i_{\max}\eta_0}\left(12\sqrt{\eta_0}L(\mathfrak{R}+\mathfrak{D})\mathfrak{R} + 2C_\xi(\mathfrak{R} + 7\eta_0 L(\mathfrak{R}+\mathfrak{D})) + \sqrt{\eta_0}C_\xi^2\right)\sqrt{\log\left(\frac{2i_{\max}}{\varepsilon_1}\right)} \tag{96}$$

$$+ 16(C_1+\mathfrak{D})^2 + i_{\max}\eta_0\left[2\eta_0(37L^2(\mathfrak{R}+\mathfrak{D})^2) + d\right] \tag{97}$$

$$\le \frac{\sqrt{80}(C_1+\mathfrak{D})}{\varepsilon_2}\left(6\sqrt{2}\varepsilon_2\mathfrak{R} + 2C_\xi\left(\mathfrak{R} + \frac{7\varepsilon_2^2}{2L\mathfrak{R}}\right) + \frac{\varepsilon_2 C_\xi^2}{\sqrt{2}L\mathfrak{R}}\right)\sqrt{\log\left(\frac{2i_{\max}}{\varepsilon_1}\right)} \tag{98}$$

$$+ 16(C_1+\mathfrak{D})^2 + \frac{40(C_1+\mathfrak{D})^2}{\varepsilon_2^2}(37\varepsilon_2^2 + d). \tag{99}$$

Let $Q = \log\left(\frac{2i_{\max}}{\varepsilon_1}\right)$. It suffices to show each of the 5 terms is at most $\frac{\mathfrak{R}^2}{5}$. Below, we use $C_\xi \le 4\sqrt{d\log\left(\frac{2i_{\max}}{\varepsilon_1}\right)}$.

$$\frac{\mathfrak{R}^2}{5} \ge 24\sqrt{10}(C_1+\mathfrak{D})\mathfrak{R}\sqrt{Q} \qquad \Leftarrow \mathfrak{R} \ge 120\sqrt{10}(C_1+\mathfrak{D})\sqrt{\log\left(\frac{2i_{\max}}{\varepsilon_1}\right)} \tag{100}$$

$$\frac{\mathfrak{R}^2}{5} \ge \frac{8\sqrt{5}(C_1+\mathfrak{D})C_\xi}{\varepsilon_2}\left(\mathfrak{R} + \frac{7\varepsilon_2}{2L\mathfrak{R}}\right)\sqrt{Q} \quad \Leftarrow \mathfrak{R}^2 \ge \frac{160\sqrt{5}(C_1+\mathfrak{D})}{\varepsilon_2}\left(\mathfrak{R} + \frac{7\varepsilon_2}{2L\mathfrak{R}}\right)\sqrt{dQ} \tag{101}$$

$$\frac{\mathfrak{R}^2}{5} \geq \frac{2\sqrt{10}(C_1 + \mathfrak{D})C_\xi^2}{L\mathfrak{R}}\sqrt{Q} \qquad\qquad \Leftarrow \mathfrak{R}^3 \geq \frac{160\sqrt{10}(C_1 + \mathfrak{D})}{L}dQ^{\frac{3}{2}} \tag{102}$$

$$\frac{\mathfrak{R}^2}{5} \geq 16(C_1 + \mathfrak{D})^2 \tag{103}$$

$$\frac{\mathfrak{R}^2}{5} \geq 40(C_1 + \mathfrak{D})^2\left(40 + \frac{d}{\varepsilon_2^2}\right) \tag{104}$$

It remains to check each of these five inequalities. First, we bound $Q$.

$$i_{\max} \leq \frac{40L^2(\mathfrak{R} + \mathfrak{D})^2\,(C_1 + \mathfrak{D})^2}{{\varepsilon_2}^4}, \tag{105}$$

$$\frac{2i_{\max}}{\varepsilon_1} \leq \frac{80L^2(\mathfrak{R} + \mathfrak{D})^2\,(C_1 + \mathfrak{D})^2}{{\varepsilon_2}^4\varepsilon_1} \tag{106}$$

$$\leq \frac{100L^2\mathfrak{R}^2\,(C_1 + \mathfrak{D})^2}{{\varepsilon_2}^4\varepsilon_1} \tag{107}$$

$$\leq \frac{10^{10}L^2(C_1 + \mathfrak{D})^4 d}{\varepsilon_2^6\varepsilon_1}\log^2\left(\max\left\{L, C_1 + \mathfrak{D}, \frac{1}{\varepsilon_1}\right\}\right) \tag{108}$$

$$\log\left(\frac{2i_{\max}}{\varepsilon_1}\right) \leq 24 + 16\log\left(\max\left\{L, C_1 + \mathfrak{D}, \frac{1}{\varepsilon_1}\right\}\right) \tag{109}$$

$$\leq 40\log\left(\max\left\{L, C_1 + \mathfrak{D}, \frac{1}{\varepsilon_1}\right\}\right) \tag{110}$$

It remains to check (100)–(104). We check (100), (101), and (102):

$$120\sqrt{10}(C_1 + \mathfrak{D})\sqrt{Q} \leq 120\sqrt{10}(C_1 + \mathfrak{D})\sqrt{40\log\left(\max\left\{L, C_1 + \mathfrak{D}, \frac{1}{\varepsilon_1}\right\}\right)} \leq \mathfrak{R} \tag{111}$$

Using $\mathfrak{R} \geq \sqrt{\frac{7\varepsilon_2}{2L}} \implies \frac{7\varepsilon_2}{2L\mathfrak{R}} \leq \mathfrak{R}$,

$$\frac{160\sqrt{5}(C_1 + \mathfrak{D})}{\varepsilon_2}\left(\mathfrak{R} + \frac{7\varepsilon_2}{2L\mathfrak{R}}\right)\sqrt{d}Q \leq \frac{320\sqrt{10}(C_1 + \mathfrak{D})\sqrt{d}\mathfrak{R}}{\varepsilon_2}40\log\left(\max\left\{L, C_1 + \mathfrak{D}, \frac{1}{\varepsilon_1}\right\}\right) \leq \mathfrak{R}^2 \tag{112}$$

$$\frac{160\sqrt{10}(C_1 + \mathfrak{D})}{L}\left(\mathfrak{R} + \frac{7\varepsilon_2}{2L\mathfrak{R}}\right)\sqrt{d}Q^{\frac{3}{2}} \leq \frac{80\sqrt{10}(C_1 + \mathfrak{D})d}{L}\left(40\log\left(\max\left\{L, C_1 + \mathfrak{D}, \frac{1}{\varepsilon_1}\right\}\right)\right)^{\frac{3}{2}} \leq \mathfrak{R}^3. \tag{113}$$

The last two inequalities (103), (104) are immediate from the definition of $\mathfrak{R}$. $\qquad\square$

## C  Overview of offline result

### C.1  Overview of offline algorithm

Similarly to the online Algorithm 2, our offline Algorithm 3 also calls the variance-reduced SGLD Algorithm 1 multiple times. In the offline setting, all functions $f_1, \ldots, f_T$ are given from the start, so there is no need to run Algorithm 1 on subsets of the functions. Instead, we run SAGA-LD on $\beta f_1, \ldots, \beta f_T$, where the *inverse temperature* $\beta$ is doubled at each epoch, from roughly $\beta = \frac{1}{T}$ to $\beta = 1$. There are logarithmically many epochs, each taking $i_{\max} = \widetilde{O}_T(1)$ Markov chain steps.

Note that we cannot just run SAGA-LD on $f_1, \ldots, f_T$. The temperature schedule is necessary because we only assume a cold start and do not assume strong convexity; in order for our variance-reduced SGLD to work, the initial starting point must be $\widetilde{O}_T(1/\sqrt{T})$ rather than $\widetilde{O}_T(1)$ away from the minimum. The temperature schedule helps us get there by roughly halving the distance to the minimum each epoch; the step sizes are also halved at each epoch. Moreover, one also cannot substitute a deterministic convex optimization algorithm for initialization in our setting, since without strong convexity, deterministic convex optimization promises a point close in function value but not Euclidean distance. In contrast, our algorithm gives, with high probability, a point close enough in Euclidean distance if Assumption 2 holds.

---

**Algorithm 3** Offline variance-reduced SGLD

---

**Input:** $T \in \mathbb{N}$ and gradient oracles for functions $f_t : \mathbb{R}^d \to \mathbb{R}$, $1 \leq t \leq T$.
**Input:** step size $\eta$, batch size $b > 0$, $i_{\max} > 0$, an initial point $\mathsf{X}^0 \in \mathbb{R}^d$
**Output:** A sample $\mathsf{X}$
  1: Set $\mathsf{X} \hookleftarrow \mathsf{X}^0$ and set $\beta = 1/T$.                     ▷ Start at a high temperature, $T$.
  2: **while** $\beta < 1$ **do**
  3:      Run Algorithm 1 with step size $\eta/\beta T$, batch size $b$, number of steps $i_{\max}$, initial point $\mathsf{X}$, and functions $\beta f_t$, $1 \leq t \leq T$.
  4:      Set $\mathsf{X} \hookleftarrow \mathsf{X}^\beta$, where $\mathsf{X}^\beta$ is the output of Algorithm 1.
  5:      $\beta \hookleftarrow \max\{2\beta, 1\}$.                     ▷ Double the temperature.
  6: **end while**
  7: Return $\mathsf{X}$.

---

### C.2  Proof overview of offline result

For the offline problem, the desired result – sampling from $\pi_T$ with TV error $\varepsilon$ using $\widetilde{O}(T) + \mathrm{poly}(d, L, C, \varepsilon^{-1}) \log_2(T)$ gradient evaluations – is known either when we assume strong convexity, or we have a warm start. We show how to achieve the same additive bound without either assumption.

Without strong convexity, we do not have access to a Lyapunov function which guarantees that the distance between the Markov chain and the mode $x^\star$ of the target distribution contracts at each step, even from a cold start. To get around this problem, we sample from a sequence of $\log_2(T)$ distributions $\pi_T^\beta \propto e^{-\beta \sum_{t=1}^T f_t(x)}$, where the inverse "temperature" $\beta$ doubles at each epoch from $\frac{1}{T}$ to 1, causing the distribution $\pi_T^\beta$ to have a decreasing second moment and to become more "concentrated" about the mode $x^\star$ at each epoch. This temperature schedule allows our algorithm to gradually approach the target distribution, even though our algorithm is initialized from a cold start $x^0$ which may be far from a sub-level set containing most of the target probability measure. The same martingale exit time argument as in the proof for the online problem shows that at the end of each epoch, the Markov chain is at a distance from $x^\star$ comparable to the (square root of the) second moment of the current distribution $\pi_T^\beta$. This provides a "warm start" for the next distribution $\pi_T^{2\beta}$, and in this way our Markov chain approaches the target distribution $\pi_T^1$ in $\log_2(T)$ epochs.

The total number of gradient evaluations is therefore $T \log_2(T) + b \times i_{\max}$, since we only compute the full gradient at the beginning of each of the $\log_2(T)$ epochs, and then only use a batch size $b$ for the gradient steps at each of the $i_{\max}$ steps of the Markov chain. As in the online case, $b$ and $i_{\max}$ are poly-logarithmic in $T$ and polynomial in the various parameters $d, L, C, \varepsilon^{-1}$, implying that the total number of gradient evaluations is $\widetilde{O}(T) + \mathrm{poly}(d, C, \mathfrak{D}, \varepsilon^{-1}, L) \log_2(T)$, in the offline setting where our goal is only to sample from $\pi_T^1$.

The proof of Theorem A.1 is similar to the proof of Theorem 2.1, except for some differences as to how the stochastic gradients are computed and how one defines the functions "$F_t$". We define $F_t := \beta_t \sum_{k=1}^T f_k$, where $\beta_t = \begin{cases} 2^{t-1}/T, & 0 \leq s \leq \log_2(T) + 1 \\ 1, & t = \lceil \log_2(T) \rceil + 1. \end{cases}$. We then show that for this choice of $F_t$ the offline assumptions, proof and algorithm are similar to those of the online case.

## D  Proof of offline theorem (Theorem A.1)

The proof of Theorem A.1 is similar to the proof of Theorem 2.1, except for some key differences as to how the stochastic gradients are computed and how one defines the functions "$F_t$".

We define $F_\beta := \beta F = \beta \sum_{k=1}^T f_k$, where the $\beta$'s will range over the sequence

$$\beta_t = \begin{cases} 2^t/T, & 0 \leq t \leq \log_2(T) \\ 1, & t = \lceil \log_2(T) \rceil. \end{cases} \tag{114}$$

For this choice of $F_\beta$, the offline assumptions, proof and algorithm are similar to those of the online case.

**Differences in assumptions.**  We have that $F_\beta$ is $\beta T L$-smooth, which (except for Lemma B.2) is the only way in which Assumption 1 is used in the proof of Theorem 2.1.

Moreover, Assumption 4 for the offline case implies that $\pi_T^\beta \propto e^{-F_\beta}$ satisfies Assumption 2 with constants $C$ and $k$ for every $t$. Since the minimizer $x_\beta^\star$ of $F_\beta$ does not change with $t$, $x_\beta^\star$ satisfies Assumption 3 with constant $\mathfrak{D} = 0$.

**Differences in algorithm.** The step size used in Algorithm 3 is $\frac{\eta}{\beta T}$, the same step size used in Algorithm 2. Thus, we note that Algorithm 3 is similar to Algorithm 2 except for a few key differences:

1. The way in which the stochastic gradient $g_i^\beta$ is computed is different. Specifically, in the offline algorithm our stochastic gradient is computed as

$$g_i^\beta = s + \frac{\beta T}{b} \sum_{k \in S} (G_{\text{new}}^k - G^k). \tag{115}$$

   where $S$ is a multiset of size $b$ chosen with replacement from $\{1, \ldots, T\}$ (rather than from $\{1, \ldots, t\}$).
2. There are logarithmically many epochs.

We now give the proof in some detail.

Letting $X_i^\beta$ be the iterates at inverse temperature $\beta$, define

$$G_\beta = \left\{ \forall i, \left\| X_i^\beta - x^\star \right\| \le \frac{\mathfrak{R}}{\sqrt{\beta T}} \right\}. \tag{116}$$

**Lemma D.1** (Analogue of Lemma B.6). *Assume that Assumptions 1 and 4 hold. Let* $C = \left( 2 + \frac{1}{k} \right) \log \left( \frac{A}{k^2} \right)$, $C_1 \ge C$, *and suppose*

$$\eta_0 \le \frac{\varepsilon_2^2}{Ld + 4L^2 \mathfrak{R}^2 / b}, \tag{117}$$

$$i_{\max} \ge \frac{5 C_1^2}{\eta_0 \varepsilon_2^2}. \tag{118}$$

*Suppose $\varepsilon_1 > 0$ is such that*

$$\mathbb{P} \left( \forall 0 \le i \le i_{\max}, \left\| X_i^\beta - x^\star \right\| \le \frac{\mathfrak{R}}{\sqrt{\beta T}} \,\middle|\, \left\| X_0^\beta - x^\star \right\| \le \frac{C_1}{\sqrt{\beta T}} \right) \ge 1 - \varepsilon_1. \tag{119}$$

*Suppose $\left\| X_0^\beta - x^\star \right\| \le \frac{2 C_1}{\sqrt{\beta T}}$. Then*

1. $\left\| \mathcal{L}(X^\beta) - \pi_T^\beta \right\|_{TV} \le \varepsilon_1 + \varepsilon_2.$

2. *For $i \in [i_{\max}]$ chosen at random,*

$$\mathbb{P} \left( \left\| X_i^\beta - x^\star \right\| \le \frac{C_1}{\sqrt{\beta T}} \right) \ge 1 - (\varepsilon_1 + \varepsilon_2 + A e^{-k C_1}). \tag{120}$$

*Proof.* First we calculate the distance of the starting point from the stationary distribution,

$$W_2^2(\delta_{X_0^\beta}, \pi_T^\beta) \le 2 \left\| X_0^\beta - x^\star \right\|^2 + 2 W_2^2(\delta_{x^\star}, \pi_T^\beta) \le \frac{8 C_1^2}{\beta T} + \frac{2 C^2}{\beta T} \le \frac{10 C_1^2}{\beta T}. \tag{121}$$

Define a toy Markov chain coupled to $X_i^\beta$ as follows. Let $\widetilde{X}_0^\beta = X_0^\beta$ and

$$\widetilde{X}_{i+1}^\beta = \begin{cases} \widetilde{X}_i^\beta - \eta g_i^\beta + \sqrt{\eta} \xi_i, & \text{when } \left\| \widetilde{X}_j^\tau - x^\star \right\| \le \frac{\mathfrak{R}}{\sqrt{\beta T}} \text{ for all } 0 \le j \le i \\ \widetilde{X}_i^\beta - \eta \beta \nabla F(\widetilde{X}_i), & \text{otherwise.} \end{cases} \tag{122}$$

By Lemma B.2, the variance of $g_i^\beta$ is at most $\frac{\beta^2 T^2 L^2}{b} \max_{0 \le j \le i} \left\| \widetilde{X}_i^\beta - \widetilde{X}_j^\beta \right\|^2$. If $\left\| X_i^\beta - x^\star \right\| \le \frac{\mathfrak{R}}{\sqrt{\beta T}}$ for all $0 \le i \le i_{\max}$, then $\left\| \widetilde{X}_i^\beta - \widetilde{X}_j^\beta \right\| \le \frac{2\mathfrak{R}}{\sqrt{\beta T}}$ for all $0 \le i, j \le i_{\max}$. Then we can apply

Lemma B.4 with $\varepsilon = 2\varepsilon_2^2$, $L \hookleftarrow L\beta T$, $\sigma^2 \leq \frac{(\beta T)^2 L^2}{b} \frac{4\mathfrak{R}^2}{\beta T} = \frac{4\beta T L^2 \mathfrak{R}^2}{b}$, and $W_2^2(\mu_0, \pi) \leq \frac{10C_1^2}{\beta T}$. By Pinsker's inequality, for random $i \in [i_{\max}]$,

$$\left\| \mathcal{L}(\widetilde{X}_i^\beta) - \pi_T^\beta \right\|_{\text{TV}} \leq \sqrt{\frac{1}{2}\text{KL}(\widetilde{\mu}|\pi_\tau)} \leq \varepsilon_2. \tag{123}$$

Under $G_\beta$, $X_i^\beta = \widetilde{X}_i^\beta$ for all $i \leq i_{\max}$ and $s \leq \tau$, so

$$\|\mathcal{L}(X_i^\beta) - \pi_T^\beta\|_{\text{TV}} \leq \mathbb{P}(G_\beta^c) + \left\| \mathcal{L}(\widetilde{X}_i^\beta) - \pi_T^\beta \right\|_{\text{TV}} \leq \varepsilon_1 + \varepsilon_2. \tag{124}$$

This shows part 1.

For part 2, note that by Assumption 2,

$$\mathbb{P}_{X \sim \pi_T^\beta}\left[ \|X - x^\star\| \geq \frac{C_1}{\sqrt{\beta T}} \right] \leq Ae^{-kC_1}. \tag{125}$$

Combining (124) and (125) gives part 2. $\qquad\square$

**Theorem D.2** (Theorem A.1 with parameters). *Suppose that Assumptions 1 and 4 hold, with $L \geq 1$, $k \leq 1$, and $\|X^0 - x^\star\| \leq C$. Suppose Algorithm 3 is run with parameters $\eta_0, i_{\max}$ given by*

$$\varepsilon_1 = \frac{\varepsilon}{3\lceil \log_2(T) + 1 \rceil}, \tag{126}$$

$$C_1 = \left( 2 + \frac{1}{k} \right) \log\left( \frac{A}{\varepsilon_2 k^2} \right), \tag{127}$$

$$\mathfrak{R} = \frac{10000C_1\sqrt{d}}{\varepsilon_1} \log\left( \max\left\{ L, C_1 + \mathfrak{D}, \frac{1}{\varepsilon_1} \right\} \right) \tag{128}$$

$$\eta_0 = \frac{\varepsilon_1^2}{2L^2\mathfrak{R}^2}, \tag{129}$$

$$i_{\max} = \left\lceil \frac{5C_1^2}{\eta_0\varepsilon_1^2} \right\rceil = \left\lceil \frac{10L^2\mathfrak{R}^2 C_1^2}{\varepsilon_1^4} \right\rceil, \tag{130}$$

*with any constant batch size $b \geq 4$. Then it outputs $X^1$ such that $X^1$ is a sample from $\widetilde{\pi}_T$ satisfying $\|\widetilde{\pi}_T - \pi_T\|_{\text{TV}} \leq \varepsilon$, using $\widetilde{O}(T) + \text{poly}\log(T)\,\text{poly}(d, L, C, \varepsilon^{-1})$ gradient evaluations.*

*proof of Theorem A.1.* The proof is similar to the proof of Theorem 2.1, and we omit the details. We show by induction that

$$\mathbb{P}\left( \left\| X_i^{\beta_s} - x^\star \right\| \leq \frac{\mathfrak{R}}{\sqrt{\beta_s T}} \right) \geq 1 - 2s\varepsilon_1. \tag{131}$$

The base case follows from $C \leq C_1 \leq \mathfrak{R}$. The induction step follows from noting first that

$$\left\| X_i^{\beta_s} - x^\star \right\| \leq \frac{\mathfrak{R}}{\sqrt{\beta_s T}} \implies \left\| X_0^{\beta_{s+1}} - x^\star \right\| \leq \frac{2\mathfrak{R}}{\sqrt{\beta_{s+1} T}}, \tag{132}$$

noting that the conditions imply (for $\eta_\beta = \frac{\eta_0}{\sqrt{\beta T}}$, $r_t = \frac{\mathfrak{R}}{\sqrt{\beta T}}$, $S_t = 4\sqrt{\beta T} L\mathfrak{R}$, and $C_\xi = \sqrt{2d + 8\log\left( \frac{2i_{\max}}{\varepsilon_1} \right)}$) that

$$\varepsilon_1 \geq i_{\max}\left[ \exp\left( -\frac{(r_\beta^2 - \frac{4C_1^2}{t+L_0/L} - i[2\eta_t^2(S_\beta^2 + L^2 t^2 r_\beta^2) + \eta_\beta d])^2}{2i_{\max}(2\eta_\beta S_\beta r_\beta + 2\sqrt{\eta_\beta}C_\xi(r_\beta + \eta_\beta S_\beta + \eta_\beta Ltr_t) + \eta_\beta C_\xi^2)^2} \right) \right. \tag{133}$$

$$\left. + \exp\left( -\frac{C_\xi^2 - d}{8} \right) \right]. \tag{134}$$

Then using Lemma B.3, we get that (119) is satisfied with $\varepsilon_1$, and the induction step follows from part 2 of Lemma D.1.

Finally, once we have $\left\| X_0^1 - x^\star \right\| \leq \frac{\mathfrak{R}}{\sqrt{T}}$, the conclusion about $X^1$ follows from part 1 of Lemma D.1. $\qquad\square$

# E Proof for logistic regression application

## E.1 Theorem for general posterior sampling, and application to logistic regression

We show that under some general conditions—roughly, that we see data in all directions—the posterior distribution concentrates. We specialize to logistic regression and show that the posterior for logistic regression concentrates under reasonable assumptions.

The proof shares elements with the proof of the Bernstein-von Mises theorem (see e.g. [Nic12]), which says that under some weak smoothness and integrability assumptions, the posterior distribution after seeing iid data (asymptotically) approaches a normal distribution. However, we only need to prove a weaker result—not that the posterior distribution is close to normal, but just $\alpha T$-strongly log concave in a neighborhood of the MLE, for some $\alpha > 0$; hence, we get good, nonasymptotic bounds. This is true under more general assumptions; in particular, the data do not have to be iid, as long as we observe data "in all directions."

**Theorem E.1** (**Validity of the assumptions for posterior sampling**). *Suppose that $\|\theta_0\| \leq B$, $x_t \sim P_x(\cdot|x_{1:t-1}, \theta_0)$. Let $f_t$, $t \geq 1$ be such that $P_x(x_t|x_{1:t-1}, \theta) \propto e^{-f_t(\theta)}$ and let $\pi_t(\theta)$ be the posterior distribution, $\pi_t(\theta) \propto e^{-\sum_{k=0}^t f_t(\theta)}$. Suppose there is $M, L, r, \sigma_{\min}, T_{\min} > 0$ and $\alpha, \beta \geq 0$ such that the following conditions hold:*

1. *For each $t$, $1 \leq t \leq T$, $f_t(\theta)$ is twice continuously differentiable and convex.*

2. *(Gradients have bounded variation) For each $t$, given $x_{1:t-1}$,*
$$\|\nabla f_t(\theta) - \mathbb{E}[\nabla f_t(\theta)|x_{1:t-1}]\| \leq M. \tag{135}$$

3. *(Smoothness) Each $f_t$ is $L$-smooth, for $1 \leq t \leq T$.*

4. *(Strong convexity in neighborhood) Let*
$$\widehat{I}_T(\theta) := \frac{1}{T} \sum_{t=1}^{T} \nabla^2 f_t(\theta). \tag{136}$$
   *Then for $T \geq T_{\min}$, with probability $\geq 1 - \frac{\varepsilon}{2}$,*
$$\forall \theta \in \mathrm{B}(\theta_0, r), \qquad \widehat{I}_T(\theta) \succeq \sigma_{\min} I_d. \tag{137}$$

5. *$f_0(\theta)$ is $\alpha$-strongly convex and $\beta$-smooth, and has minimum at $\theta = 0$.*

*Let $\theta_T^\star$ be the minimum of $\sum_{t=0}^{T} f_t(\theta)$, i.e., the mode for $\theta$ after observing $x_{1:T}$. Letting*
$$C = \max\left\{1, M\sqrt{2d\log\left(\frac{2d}{\varepsilon}\right)}, \frac{4d}{\sigma_{\min}}\right\},$$
*and $c = \frac{\alpha}{\sigma_{\min}}$, if $T \geq T_{\min}$ is such that $\frac{C\sqrt{T}+\beta B}{\sigma_{\min}T+\alpha} + \frac{C}{\sqrt{T+c}} < r$, then with probability $1 - \varepsilon$, the following hold:*

1. *$\|\theta_T^\star - \theta_0\| \leq \frac{C\sqrt{T}+\beta B}{\sigma_{\min}T+\alpha}$.*

2. *For $C' \geq 0$, $\mathbb{P}_{\theta \sim \pi_T}\left(\|\theta - \theta_T^\star\| \geq \frac{C'}{\sqrt{T+c}}\right) \leq \frac{K_1}{\sigma_{\min}C\sqrt{T+c}}\left(\frac{(LT+\beta)e}{d}\right)^{\frac{d}{2}} e^{\frac{1}{2}\sigma_{\min}C^2 - \frac{\sigma_{\min}CC'}{2}}$ for some constant $K_1$.*

The strong convexity condition is analogous to a small-ball inequality [KM15, Men14] for the sample Fisher information matrix in a neighborhood of the true parameter value. In the iid case we have concentration (which is necessary for a central limit theorem to hold, as in the Bernstein-von Mises Theorem); in the non-iid case we do not necessarily have concentration, but the small-ball inequality can still hold.

We show that under reasonable conditions on the data-generating distribution, logistic regression satisfies the above conditions. Let $\phi(x) = \frac{1}{1+e^{-x}}$ be the logistic function. Note that $\phi(-x) = 1 - \phi(x)$.

Applying Theorem E.1 to the setting of logistic regression, we will obtain the following.

**Lemma E.2.** *In the setting of Problem 2.2 (logistic regression), suppose that $\|\theta_0\| \leq \mathfrak{B}$, $u_t \sim P_u$ are iid, where $P_u$ is a distribution that satisfies the following: for $u \sim P_u$,*

1. *(Bounded) $\|u\|_2 \leq M$ with probability 1.*

2. *(Minimal eigenvalue of Fisher information matrix)*

$$I(\theta_0) := \int_{\mathbb{R}^d} \phi(u^\top \theta_0)\phi(-u^\top \theta_0) uu^\top \, dP_u \succeq \sigma I_d, \tag{138}$$

   *for $\sigma > 0$.*

*Let*

$$C = \max \left\{ 1, 2M\sqrt{2d\log\left(\frac{2d}{\varepsilon}\right)}, \frac{4ed}{\sigma} \right\}. \tag{139}$$

*Then for $t > \max\left\{ \frac{M^4 \log\left(\frac{2d}{\varepsilon}\right)}{8\sigma^2}, 4M^2\left(\frac{2eC}{\sigma}+1\right)^2, \frac{4eM\mathfrak{B}\alpha}{\sigma} \right\}$, we have*

1. *$\nabla f_k(\theta)$ is $\frac{M^2}{4}$-Lipschitz for all $k \in \mathbb{N}$.*

2. *For any $C' \geq 0$, and $c = \frac{2e\alpha}{\sigma}$,*

$$\mathbb{P}_{\theta \sim \pi_t}\left(\|\theta - \theta_t^\star\| \geq \frac{C'}{\sqrt{T+c}}\right) \leq \frac{K_1}{\sigma C \sqrt{T+c}} \left(\frac{\left(\frac{M^2}{4}T + \alpha\right)e}{d}\right)^{\frac{d}{2}} e^{\frac{1}{4e}\sigma C^2 - \frac{\sigma C C'}{4e}} \tag{140}$$

   *for some constant $K_1$.*

3. *With probability $1 - \varepsilon$, $\|\theta_t^\star - \theta_0\| \leq \frac{C\sqrt{t}+\alpha\mathfrak{B}}{\sigma t/2e+\alpha}$.*

**Remark E.3.** *We explain the condition $I(\theta_0) = \int_{\mathbb{R}^d} \phi(u^\top \theta_0)\phi(-u^\top \theta_0) uu^\top \, dP_u \succeq \sigma I_d$. Note that $\phi(x)\phi(-x)$ can be bounded away from 0 in a neighborhood of $x = 0$, and then decays to 0 exponentially in $x$. Thus, $I(\theta_0)$ is essentially the second moment, where we ignore vectors that are too large in the direction of $\pm\theta_0$.*

*More precisely, we have the following implication:*

$$\mathbb{E}_u[uu^\top \mathbb{1}_{\phi(u^\top \theta_0) \leq C_1}] \succeq \sigma I_d \implies \int_{\mathbb{R}^d} \phi(u^\top \theta_0)\phi(-u^\top \theta_0) uu^\top \, dP_u \succeq \frac{1}{\phi(C_1)(1 - \phi(C_1))}\sigma I_d. \tag{141}$$

*Theorem 2.3 is stated with $C_1 = 2$.*

## E.2 Proof of Theorem E.1

*Proof of Theorem E.1.* Let $\mathcal{E}$ be the event that (137) holds.

Step 1: We bound $\|\theta_T^\star - \theta_0\|$ with high probability.

We show that with high probability $\sum_{t=0}^T \nabla f_t(\theta_0)$ is close to 0. Since $\sum_{t=0}^T \nabla f_t(\theta_T^\star) = 0$, the gradient at $\theta_0$ and $\theta_T^\star$ are close. Then by strong convexity, we conclude $\theta_0$ and $\theta_T^\star$ are close.

First note that $\mathbb{E}[f_t(\theta)|x_{1:t-1}] = \int_{\mathbb{R}^d} -\log P_x(x_t|x_{1:t-1}, \theta) \, dP_x(\cdot|x_{1:t-1}, \theta_0)$ is a KL divergence minus the entropy for $P_x(\cdot|x_{1:t-1}, \theta_0)$, and hence is minimized at $\theta = \theta_0$. Hence $\frac{1}{T}\sum_{t=1}^T \mathbb{E}[\nabla f_t(\theta_0)|x_{1:t-1}] = 0$. Thus by Lemma I.1 applied to

$$\sum_{t=1}^T \nabla f_t(\theta_0) = \sum_{t=1}^T \left[\nabla f_t(\theta_0) - \mathbb{E}[\nabla f_t(\theta_0)|x_{1:t-1}]\right], \tag{142}$$

we have by Chernoff's inequality that

$$\mathbb{P}\left(\left\|\sum_{t=1}^{T}\nabla f_t(\theta_0)\right\| \geq \frac{C}{\sqrt{T}}\right) \leq 2de^{-\frac{C^2}{2M^2d}} \leq \frac{\varepsilon}{2} \tag{143}$$

when $\frac{C^2}{2M^2d} \geq \log\left(\frac{4d}{\varepsilon}\right)$, which happens when $C \geq M\sqrt{2d\log\left(\frac{4d}{\varepsilon}\right)}$.

Let $\mathcal{A}$ be the event that $\left\|\frac{1}{T}\sum_{t=1}^{T}\nabla f_t(\theta_0)\right\| < \frac{C}{\sqrt{T}}$. Then under $\mathcal{A}$,

$$\left\|\frac{1}{T}\sum_{t=0}^{T}\nabla f_t(\theta_0)\right\| > -\frac{C}{\sqrt{T}} - \frac{1}{T}\beta\|\theta_0\| \geq -\frac{C}{\sqrt{T}} - \frac{\beta B}{T}. \tag{144}$$

Let $w = \frac{\theta_T^\star - \theta_0}{\|\theta_T^\star - \theta_0\|}$. Under the event $\mathcal{E}$,

$$\frac{1}{T}\sum_{t=0}^{T}\nabla f_t(\theta_0 + sw)^\top w \geq -\frac{C}{\sqrt{T}} - \frac{\beta B}{T} + \left(\sigma_{\min} + \frac{\alpha}{T}\right)\min\{s,r\}. \tag{145}$$

Hence, if $s, r > \frac{C\sqrt{T} + \beta B}{\sigma_{\min}T + \alpha}$, then $\sum_{t=0}^{T}\nabla f_t(\theta_0) \neq 0$. Considering $s = \|\theta_T^\star - \theta_0\|$, this means that

$$\|\theta_T^\star - \theta_0\| \leq \frac{C\sqrt{T} + \beta B}{\sigma_{\min}T + \alpha}. \tag{146}$$

Step 2: For $c = \frac{\alpha}{\sigma_{\min}}$, we bound $\mathbb{P}_{\theta \sim \pi_T}(\|\theta - \theta_T^\star\| \geq \frac{C'}{\sqrt{T+c}})$.

Under $\mathcal{E}$, $\frac{1}{T}\sum_{t=1}^{T}f_t(\theta)$ is $\sigma_{\min}$-strongly convex for $\theta \in \mathrm{B}\left(\theta_T^\star, \frac{C}{\sqrt{T+c}}\right) \subset \mathrm{B}(\theta_0, r)$, and $f_0(\theta)$ is $\alpha$-strongly convex.

Let $r' = r - \frac{C\sqrt{T} + \beta B}{\sigma_{\min}T + \alpha}$. Under $\mathcal{A}$, $\mathrm{B}(\theta_T^\star, r') \subset \mathrm{B}(\theta_0, r)$. Thus under $\mathcal{E} \cap \mathcal{A}$, letting $w(\theta) := \frac{\theta - \theta_T^\star}{\|\theta - \theta_T^\star\|}$,

$$\forall \theta \in \mathrm{B}(\theta_T^\star, r') \subset \mathrm{B}(\theta_0, r), \qquad \sum_{t=0}^{T}\nabla f_t(\theta)^\top w(\theta) \geq (T\sigma_{\min} + \alpha)\|\theta - \theta_T^\star\|. \tag{147}$$

Suppose $T$ is such that $\frac{C}{\sqrt{T+c}} < r'$, i.e., $\frac{C\sqrt{T} + \beta B}{\sigma_{\min}T + \alpha} + \frac{C}{\sqrt{T+c}} < r$. By shifting, we may assume that $\sum_{t=0}^{T}f_t(\theta_T^\star) = 0$. Because $f_t(\theta)$ is $L$-smooth for $1 \leq t \leq T$ and $\beta$-smooth for $t = 0$,

$$\sum_{t=0}^{T}f_t(\theta) \leq \frac{LT + \beta}{2}\|\theta - \theta_T^\star\|^2. \tag{148}$$

Then for all $\theta \in \mathrm{B}\left(\theta_T^\star, \frac{C}{\sqrt{T+c}}\right)^c$,

$$\sum_{t=0}^{T}f_t(\theta) \geq \sum_{t=0}^{T}f_t\left(\theta_T^\star + \frac{C}{\sqrt{T+c}}w(\theta)\right) + \sum_{t=0}^{T}\left[f_t(\theta) - f_t\left(\theta_T^\star + \frac{C}{\sqrt{T+c}}w(\theta)\right)\right] \tag{149}$$

$$\geq \frac{1}{2}(T\sigma_{\min} + \alpha)\frac{C^2}{T+c} + (T\sigma_{\min} + \alpha)\frac{C}{\sqrt{T+c}}\left(\|\theta - \theta_T^\star\| - \frac{C}{\sqrt{T+c}}\right) \tag{150}$$

$$\geq \frac{1}{2}\sigma_{\min}C^2 + \sigma_{\min}C\sqrt{T+c}\left(\|\theta - \theta_T^\star\| - \frac{C}{\sqrt{T+c}}\right). \tag{151}$$

Thus for any $C' \geq 0$,

$$\int_{\mathbb{R}^d} e^{-\sum_{t=0}^T f_t(\theta)} \, d\theta \geq \int_{\mathbb{R}^d} e^{-\frac{LT+\beta}{2}\|\theta-\theta_T^\star\|^2} \, d\theta = \left(\frac{2\pi}{LT+\beta}\right)^{\frac{d}{2}}, \tag{152}$$

$$\int_{B\left(\theta_T^\star, \frac{C'}{\sqrt{T+c}}\right)^c} e^{-\sum_{t=0}^T f_t(\theta)} \, d\theta \leq \int_{B\left(\theta_T^\star, \frac{C'}{\sqrt{T+c}}\right)^c} e^{-\frac{1}{2}\sigma_{\min}C^2} e^{-\sigma_{\min}C\sqrt{T+c}\left(\|\theta-\theta_T^\star\|-\frac{C}{\sqrt{T+c}}\right)} \, d\theta \tag{153}$$

$$= \int_{\frac{C'}{\sqrt{T+c}}}^\infty \mathrm{Vol}_{d-1}(\mathbb{S}^{d-1})\gamma^{d-1} e^{\frac{1}{2}\sigma_{\min}C^2} e^{-\sigma_{\min}C\sqrt{T+c}\gamma} \, d\gamma \tag{154}$$

$$= \int_{\frac{C'}{\sqrt{T+c}}}^\infty \mathrm{Vol}_{d-1}(\mathbb{S}^{d-1}) e^{\frac{1}{2}\sigma_{\min}C^2} e^{-(\sigma_{\min}C\sqrt{T+c}\gamma-(d-1)\log\gamma)} \, d\gamma. \tag{155}$$

Now, when $C \geq \max\{\frac{2(d-1)}{\sigma_{\min}}, 1\}$, we have that

$$\sigma_{\min}C\sqrt{T+c}\gamma - (d-1)\log\gamma \geq \sigma_{\min}C\sqrt{T+c}\gamma - (d-1)\gamma \tag{156}$$

$$\geq \sigma_{\min}C\sqrt{T+c}\gamma - \frac{\sigma_{\min}C\sqrt{T+c}\gamma}{2} \tag{157}$$

$$= \frac{\sigma_{\min}C\sqrt{T+c}\gamma}{2}. \tag{158}$$

Then by Stirling's formula, for some $K_1$,

$$(155) \leq \mathrm{Vol}_{d-1}(\mathbb{S}^{d-1}) e^{\frac{1}{2}\sigma_{\min}C^2} \int_{\frac{C'}{\sqrt{T+c}}}^\infty e^{-\frac{\sigma_{\min}C\sqrt{T+c}\gamma}{2}} \, d\gamma \tag{159}$$

$$\leq \frac{2\pi^{\frac{d}{2}}}{\Gamma\left(\frac{d}{2}\right)} e^{\frac{1}{2}\sigma_{\min}C^2} \frac{2}{\sigma_{\min}C\sqrt{T+c}} e^{-\frac{\sigma_{\min}CC'}{2}} \tag{160}$$

$$\leq \frac{K_1}{\sigma_{\min}C\sqrt{T+c}} \left(\frac{2\pi e}{d}\right)^{\frac{d}{2}} e^{\frac{1}{2}\sigma_{\min}C^2 - \frac{\sigma_{\min}CC'}{2}}. \tag{161}$$

We bound $\mathbb{P}_{\theta\sim\pi_T}\left(\|\theta-\theta_T^\star\| \geq \frac{C'}{\sqrt{T+c}}\right)$. By (152) and (155),

$$\mathbb{P}_{\theta\sim\pi_T}\left(\|\theta-\theta_T^\star\| \geq \frac{C'}{\sqrt{T+c}}\right) = \frac{\int_{\theta\in B\left(\theta_T^\star, \frac{C'}{\sqrt{T+c}}\right)^c} e^{-\sum_{t=0}^T f_t(\theta)} \, d\theta}{\int_{\mathbb{R}^d} e^{-\sum_{t=0}^T f_t(\theta)} \, d\theta} \tag{162}$$

$$\leq \frac{K_1}{\sigma_{\min}C\sqrt{T+c}} \left(\frac{LT+\beta}{2\pi}\right)^{\frac{d}{2}} \left(\frac{2\pi e}{d}\right)^{\frac{d}{2}} e^{\frac{1}{2}\sigma_{\min}C^2 - \frac{\sigma_{\min}CC'}{2}} \tag{163}$$

$$= \frac{K_1}{\sigma_{\min}C\sqrt{T+c}} \left(\frac{(LT+\beta)e}{d}\right)^{\frac{d}{2}} e^{\frac{1}{2}\sigma_{\min}C^2 - \frac{\sigma_{\min}CC'}{2}}, \tag{164}$$

as needed. The requirements on $C$ are $C \geq \max\left\{1, M\sqrt{2d\log\left(\frac{4d}{\varepsilon}\right)}, \frac{2d}{\sigma_{\min}}\right\}$, so the theorem follows. $\qquad\square$

## E.3 Online logistic regression: Proof of Lemma E.2 and Theorem 2.3

To prove Lemma E.2, we will apply Theorem E.1. To do this, we need to verify the conditions in Theorem E.1.

**Lemma E.4.** *Under the assumptions of Lemma E.2,*

1. *(Gradients have bounded variation) For all $t$, $\|\nabla f_t(\theta)\| \leq M$ and $\|\nabla f_t(\theta) - \mathbb{E}\nabla f_t(\theta)\| \leq 2M$.*

2. *(Smoothness) For all $t$, $f_t$ is $\frac{1}{4}M^2$-smooth.*

3. *(Strong convexity in neighborhood) for $T \geq \frac{M^4 \log\left(\frac{d}{\varepsilon}\right)}{8\sigma^2}$,*

$$\mathbb{P}\left(\forall \theta \in \mathrm{B}\left(\theta_0, \frac{1}{M}\right), \sum_{t=1}^{T} \nabla^2 f_t(\theta) \succeq \frac{\sigma}{2e}TI_d\right) \geq 1 - \varepsilon. \tag{165}$$

*Proof.* First, we calculate the Hessian of the negative log-likelihood.

If $f_t(\theta) = -\log \phi(yu^\top\theta)$, then

$$\nabla f_t(\theta) = \frac{-y\phi(yu^\top\theta)\phi(-yu^\top\theta)}{\phi(yu^\top\theta)}u = -y\phi(-yu^\top\theta)u, \tag{166}$$

$$\nabla^2 f_t(\theta) = \phi(-yu^\top\theta)\phi(yu^\top\theta)uu^\top. \tag{167}$$

Note that $\|\nabla f_t(\theta)\| \leq \|u\| \leq M$, so the first point follows.

To obtain the expected values, note that $y = 1$ with probability $\phi(u^\top\theta_0)$, and $y = -1$ with probability $1 - \phi(u^\top\theta_0)$, so that

$$\mathbb{E}[\nabla^2 f_t(\theta)] = \mathbb{E}_{(u,y)}[\phi(-yu^\top\theta)\phi(yu^\top\theta)uu^\top] \tag{168}$$

$$= \mathbb{E}_u[\phi(u^\top\theta_0)\phi(-yu^\top\theta)\phi(yu^\top\theta)uu^\top + (1 - \phi(u^\top\theta_0))\phi(-yu^\top\theta)\phi(yu^\top\theta)uu^\top] \tag{169}$$

$$= \mathbb{E}_u[\phi(u^\top\theta)(1 - \phi(u^\top\theta))uu^\top]. \tag{170}$$

Suppose that $\mathbb{E}_u[\phi(u^\top\theta)(1 - \phi(u^\top\theta))uu^\top] \succeq \sigma I$.

Next, we show that $\sum_{t=1}^{T} \nabla^2 f_t(\theta_0)$ is lower-bounded with high probability.

Note that $\left\|\nabla^2 f_t(\theta_0)\right\| = \left\|\phi(-yu^\top\theta_0)\phi(yu^\top\theta_0)uu^\top\right\|_2 \leq \frac{1}{4}M^2$. (So the second point follows.) By the Matrix Chernoff bound,

$$\mathbb{P}\left(\sum_{t=1}^{T} \nabla f_t^2(\theta_0) \not\succeq \frac{\sigma}{2}TI_d\right) \leq de^{-\frac{2\cdot 4^2}{M^4}T\left(\frac{\sigma}{2}\right)^2} = de^{-\frac{8\sigma^2 T}{M^4}} \leq \varepsilon, \tag{171}$$

when $T \geq \frac{M^4 \log\left(\frac{d}{\varepsilon}\right)}{8\sigma^2}$.

Finally, we show that if the minimum eigenvalue of this matrix is bounded away from 0 at $\theta_0$, then it is also bounded away from 0 in a neighborhood. To see this, note

$$\frac{\phi(x+c)(1-\phi(x+c))}{\phi(x)(1-\phi(x))} = \frac{e^{x+c}}{(1+e^{x+c})^2}\frac{(1+e^x)^2}{e^x} \geq \frac{e^c}{e^{2c}} = e^{-c}. \tag{172}$$

Therefore, if $\sum_{t=1}^{T} \nabla^2 f_t(\theta_0) \succeq \sigma'I_d$, then for $\|\theta - \theta_0\|_2 \leq \frac{1}{M}$, $|u^\top\theta - u^\top\theta_0| < 1$ so by (172),

$$\sum_{t=1}^{T} \nabla^2 f_t(\theta) = \sum_{t=1}^{T} \phi(u_t^\top\theta)(1 - \phi(u_t^\top\theta))u_t u_t^\top \tag{173}$$

$$\succeq \sum_{t=1}^{T} e^{-1}\phi(u_t^\top\theta_0)(1 - \phi(u_t^\top\theta_0))u_t u_t^\top \succeq \frac{\sigma'}{e}I_d. \tag{174}$$

Therefore,

$$\mathbb{P}\left(\forall \theta \in \mathrm{B}\left(\theta_0, \frac{1}{M}\right), \sum_{t=1}^{T} \nabla^2 f_t(\theta) \not\succeq \frac{\sigma}{2e}TI_d\right) \leq \mathbb{P}\left(\sum_{t=1}^{T} \nabla f_t^2(\theta_0) \not\succeq \frac{\sigma}{2}TI_d\right) \leq \varepsilon. \tag{175}$$

$\square$

*Proof of Lemma E.2.* Part 1 was already shown in Lemma E.4.

Lemma E.4 shows that the conditions of Theorem E.1 are satisfied with $M \leftarrow 2M$, $L = \frac{M^2}{4}$, $r = \frac{1}{M}$, $\sigma_{\min} = \frac{\sigma}{2e}$, $T_{\min} = \frac{M^4 \log\left(\frac{2d}{\varepsilon}\right)}{8\sigma^2}$. Also, $\alpha = \beta$. We further need to check that the condition on $t$ implies that $\frac{C\sqrt{t}+\beta\mathfrak{B}}{\sigma_{\min}t+\alpha} + \frac{C}{\sqrt{t}} < \frac{1}{M}$. We have, noting $\sigma_{\min} \leq L$ (the strong convexity is at most the smoothness),

$$\frac{C\sqrt{t} + \beta\mathfrak{B}}{\sigma_{\min}t + \alpha} + \frac{C}{\sqrt{t}} \leq \left(\frac{C}{\sigma_{\min}} + 1\right)\frac{1}{\sqrt{t + \frac{\alpha}{L}}} + \frac{\beta\mathfrak{B}}{\sigma_{\min}\left(t + \frac{\alpha}{\sigma_{\min}}\right)}, \tag{176}$$

so it suffices to have each entry be $< \frac{1}{2M}$, and this holds when $t > 4M^2\left(\frac{C}{\sigma_{\min}} + 1\right)^2 = 4M^2\left(\frac{2eC}{\sigma} + 1\right)^2$ and $t > \frac{2M\mathfrak{B}\beta}{\sigma_{\min}} = \frac{4eM\mathfrak{B}\alpha}{\sigma}$.

Parts 2 and 3 then follow immediately. □

*Proof of Theorem 2.3.* Redefine $\sigma$ such that $I(\theta_0) \succeq \sigma I_d$ holds. (By Remark E.3, this $\sigma$ is a constant factor times the $\sigma$ in Theorem 2.3) Theorem 2.3 follows from Theorem 2.1 once we show that Assumptions 1, 2, and 3 are satisfied. Assumption 1 is satisfied with $L_0 = \alpha$ and $L = \frac{M^2}{4}$. The rest will follow from Lemma E.2 except that we need bounds to cover the case $t \leq T_{\min} := \max\left\{\frac{M^4 \log\left(\frac{2d}{\varepsilon}\right)}{8\sigma^2}, \frac{16e^2M^2C^2}{\sigma^2}, \frac{4eM\mathfrak{B}\alpha}{\sigma}\right\}$ as well.

**Showing that Assumption 2 holds.** Note $L \geq \sigma$ so $\frac{C'}{\sqrt{T + \frac{\alpha}{L}}} \geq \frac{C'}{\sqrt{T + \frac{2e\alpha}{\sigma}}}$. For $t > T_{\min}$, part 2 of Lemma E.2 shows Assumption 2 is satisfied with $c = \frac{\alpha}{L}$ (where $L = \frac{M^2}{4}$), $A_1 = \frac{K_1}{\sigma C}\left(\frac{\left(\frac{M^2}{4}T+\alpha\right)e}{d}\right)^{\frac{d}{2}} e^{\frac{1}{4e}\sigma C^2}$ and $k_1 = \frac{\sigma C}{4e}$.

For $t \leq T_{\min}$, we use Lemma F.10 of [GLR18], which says that if $p(x) \propto e^{-f(x)}$ in $\mathbb{R}^d$ and $f$ is $\kappa$-strongly convex and $K$-smooth, and $x^\star = \text{argmin}_x f(x)$, then

$$\mathbb{P}_{x\sim p}\left(\|x - x^\star\|^2 \geq \frac{1}{\kappa}\left(\sqrt{d} + \sqrt{2t + d\log\left(\frac{K}{\kappa}\right)}\right)^2\right) \leq e^{-t}. \tag{177}$$

In our case, $\sum_{s=0}^{t} f_s(x)$ is $\alpha$-strongly convex and $\alpha + T_{\min}L$-smooth, so

$$\mathbb{P}_{x\sim p}\left(\|x - x^\star\| \geq \gamma\right) \leq \exp\left[-\left[\frac{(\gamma\sqrt{\kappa} - \sqrt{d})^2 - d\log\left(\frac{K}{\kappa}\right)}{2}\right]\right] \tag{178}$$

$$= e^{\frac{d}{2}\left(-1+\log\left(\frac{K}{\kappa}\right)\right)}e^{\gamma\sqrt{\kappa d}-\frac{\gamma^2\kappa}{2}} \tag{179}$$

$$\leq e^{\frac{d}{2}\left(-1+\log\left(\frac{K}{\kappa}\right)\right)-\left(\gamma-2\sqrt{\frac{d}{\kappa}}\right)\sqrt{\kappa d}}. \tag{180}$$

Thus for $t \leq T_{\min}$,

$$\mathbb{P}_{\theta\sim\pi_t}\left(\|\theta - \theta_t^\star\| \geq \gamma\right) \leq A_2 e^{-k_2\gamma} \tag{181}$$

$$\text{with } A_2 = e^{\frac{d}{2}\left(-1+\log\left(\frac{K}{\kappa}\right)\right)} = e^{\frac{d}{2}\left(-1+\log\left(\frac{T_{\min}L+\alpha}{\alpha}\right)\right)} \tag{182}$$

$$k_2 = \frac{\sqrt{\kappa d}}{\sqrt{T_{\min} + \frac{\alpha}{L}}} = \frac{\sqrt{\alpha d}}{\sqrt{T_{\min} + \frac{\alpha}{L}}}. \tag{183}$$

Take $A = \max\{A_1, A_2\}$ and $k = \min\{k_1, k_2\}$ and note that $\log(A)$, $k^{-1}$ are polynomial in all parameters and $\log(T)$.

**Showing that Assumption 3 holds.** For $t > T_{\min}$, part 3 of Lemma E.2 shows that with probability at least $1 - \varepsilon$, (using $L \geq \sigma$)

$$\|\theta_t^\star - \theta_0\| \leq \frac{C\sqrt{t} + \alpha\mathfrak{B}}{\sigma t/2e + \alpha} \leq \left(\frac{C}{\sigma/2e} + \frac{\alpha\mathfrak{B}}{\sigma/2e \cdot \sqrt{t + \frac{2e\alpha}{\sigma}}}\right) \frac{1}{\sqrt{t + \frac{\alpha}{L}}}. \tag{184}$$

Now consider $t \leq T_{\min}$. Since $F_t$ is strongly convex, the minimizer $\theta_t^\star$ of $F_t$ is the unique point where $\nabla F_t(\theta_t^\star) = 0$. Moreover, $\|\sum_{k=1}^t \nabla f_k(\theta)\| \leq T_{\min}M$ for $t \leq T_{\min}$. Therefore, since $f_0$ is $\alpha$-strongly convex, we have that $\|\nabla F_t(\theta)\| = \left\|\nabla f_0(\theta) + \sum_{k=1}^t \nabla f_k(\theta)\right\| > 0$ for all $\|\theta\| > T_{\min}M\alpha^{-1}$. Therefore, we must have that $\|\theta_t^\star\| \leq T_{\min}M\alpha^{-1}$ for all $t \leq T_{\min}$, and hence that

$$\|\theta_t^\star - \theta_0\| \leq T_{\min}M\alpha^{-1} + \mathfrak{B} \qquad \forall t \leq T_{\min}. \tag{185}$$

Set $\mathfrak{D} = 2\max\left\{(T_{\min}M\alpha^{-1} + \mathfrak{B})\sqrt{T_{\min} + \frac{\alpha}{L}}, \frac{C}{\sigma/2e} + \frac{\sqrt{\alpha}\mathfrak{B}}{\sqrt{\sigma/2e}}\right\}$. Then Equations (184) and (185) and the triangle inequality would imply that if $t < \tau$, then $\|\theta_t^\star - \theta_\tau^\star\| \leq \frac{\mathfrak{D}}{\sqrt{t + \frac{\alpha}{L}}}$. To get Assumption 3 to hold with probability at least $1 - \varepsilon$ for all $t, \tau < T$, substitute $\varepsilon \hookleftarrow \frac{\varepsilon}{T}$. $\mathfrak{D}$ is polynomial in all parameters and $\log(T)$. $\square$

# F  Discussion and future work

**Comparison to using a regularizer.**  Recall that one issue in proving Theorem 2.1 is that we don't assume the $f_t$ are strongly convex. One way to get around this is to add a strongly convex regularizer, and use existing results for Langevin in the strongly convex case. In the online case, one would have to add $\varepsilon t\|x - \hat{x}_t\|^2$ to the objective, where $\hat{x}_t$ is an estimate of the mode $x_t^\star$. Assuming we have such an estimate, using results on Langevin for strong convexity, to get $\varepsilon$ TV-error, we also require $\widetilde{O}\left(\frac{1}{\varepsilon^6}\right)$ steps per iteration. (Specifically, use [DMM19, Corollary 22], with strong convexity $m = \varepsilon t$ to get that $\widetilde{O}\left(\frac{1}{\varepsilon^3}\right)$ iterations are required to get KL-error $\varepsilon$, and apply Pinsker's inequality.)

**Preconditioning.**  Note our result does not hold if the covariance matrix of the $u_t$'s distribution becomes much more ill-conditioned over time, as is the case in certain Thompson sampling applications [RVRK+18].

We would like to obtain similar bounds under more general assumptions where the covariance matrix could change at each epoch and be ill-conditioned. This type of distribution arises in reinforcement learning applications such as Thompson sampling [DFE18], where the data is determined by the user's actions. If the user favors actions in certain "optimal" directions, in some cases the distribution may have a much smaller covariance in those directions than in other directions, causing the covariance matrix of the target distribution to become more ill-conditioned over time.

**Improved bounds for strongly convex functions.**  Suppose that we dropped the requirement of independence. Note that if we use SAGA-LD with the last sample from the previous epoch, we have a warm start for the previous distribution, and would be able to achieve TV error that decreases as $T$ with $\widetilde{O}_T(1)$ time per epoch. It seems possible to reduce the TV error to $O\left(\frac{\varepsilon}{t^{\frac{1}{6}}}\right)$ this way, and possibly to $O\left(\frac{\varepsilon}{t^{\frac{1}{4}}}\right)$ with stronger drift assumptions. These guarantees may also extend to subexponential distributions.

**Distributions over discrete spaces.**  There has been work on stochastic methods in the setting of discrete variables [DSCW18] that could potentially be used to develop analogous theory in the discrete case.

**Non-compact distributions**  One can also consider the problem of sampling from log-densities which are a sum of $T$ functions with compact support (online sampling from such distributions was considered in [NR17], but their running time bounds are not logarithmic in $T$ at each epoch). One cannot directly apply our results to compactly supported log-densities, since they do not satisfy

our Lipschitz gradient assumption (Assumption 1). At the very least we would have to modify our algorithm, for example by rejecting steps proposed by our algorithm that would otherwise cause the Markov chain to leave the support of the target distribution. A more challenging issue would be that restricting the distribution to a compact support can cause the distribution's covariance matrix to become increasingly ill-conditioned as the number of functions $t$ increases, even if the support is convex. To get around this problem we would need to modify our algorithm by including an adaptive pre-conditioner which changes along with the changing target distribution.

**Necessity of drift condition (Assumption 3).**    Since we do not assume that the individual functions $f_k$ are strongly convex, the mode (or, alternatively, the mean) of the target distribution cannot be controlled by the mode (or mean) of the individual functions. For instance, in logistic regression, all of the individual functions have "mode" at $\pm\infty$ in the direction of the data vector. Therefore, unlike in the strongly convex case, a condition on the mode of each individual function $f_k$ does not suffice for many non-strongly convex applications including logistic regression. Rather, the mode depends on the probability distribution from which the individual functions are drawn. We show that Assumption 3 holds in Section 2.4 for the special case of Bayesian logistic regression, and give more general conditions for when Assumption 3 holds in Theorem E.1.

# G    A simple example where our assumptions hold

As a simple example to motivate our assumptions, we consider the Bayesian linear regression model $y_t = z_t^\top \theta_0 + w_t$, where $y_t \in \mathbb{R}^1$ is the dependent variable, $z_t \in \mathbb{R}^d$ the independent variable, and $w_t \sim N(0,1)$ the unknown noise term. The Bayesian posterior distribution for the coefficient $\theta_0$ is $\pi_t(\theta) \propto e^{-\sum_{k=1}^t f_k(\theta)} = e^{-[\theta-\mu]^\top \Sigma^{-1}[\theta-\mu]}$ where $f_k(\theta) = (y_k - z_k\theta)^2$ for each $k$, $\Sigma^{-1} = \sum_{k=1}^T z_k z_k^\top$ and $\mu = \Sigma^{1/2} \sum_{k=1}^T y_k z_k$. Hence, the posterior $\pi_t$ has distribution $N(\mu, \Sigma)$. While computing $\Sigma$ requires at least $T \times d^2$, computing a stochastic gradient with batch size $b$ requires $d \times b$ operations. Therefore, one can hope to sample in fewer than $T \times d^2$ operations (we prove this in Theorem 2.1).

We now show that our assumptions hold for this example. For simplicity, we assume that the dimension $d = 1$, $z_t = 1$ for all $t$, and assume an improper "flat" prior, that is, $f_0 = 0$. At each epoch $t \in \{1, \dots, T\}$, the Bayesian posterior distribution for the coefficient $\theta_0$ is $\pi_t(\theta) \propto e^{-\sum_{k=1}^t f_k(\theta)}$, which a simple computation shows is the normal distribution with mean $\theta_0 + \frac{\sum_{k=0}^t w_k}{t}$ and variance $\frac{1}{2t} \le \frac{1}{t+1}$. Thus, Assumption 1 is satisfied with $L = 1$ and Assumption 2 is satisfied with $C = 2$. To verify Assumption 3, we note that $x_t^\star = \frac{\sum_{k=1}^t w_k}{t}$, and thus $x_t^\star \sim N(0, \frac{1}{t})$. We can then apply Gaussian concentration inequalities to show that $\mathfrak{D} = 4\log^{\frac{1}{2}}(\frac{\log(T)}{\delta})$ with probability at least $1 - \delta$.

# H    Hardness

**Hardness of optimization with stochastic gradients.**    The authors of [AWBR09] consider the problem of optimizing an $L$-Lipschitz function $F : \mathcal{K} \to \mathbb{R}$ on a convex body $K$ contained in an $\ell_\infty$ ball of radius $r > 0$. Given an initial point in $\mathcal{K}$ and access to a first-order stochastic gradient oracle with variance $\sigma^2$, they show that any optimization method, given a worst-case initial point in $\mathcal{K}$, requires at least $\Omega(\frac{L^2\sigma^2 d}{\delta^2})$ calls to the stochastic gradient oracle to obtain a random point $\hat{x}$ such that $\mathbb{E}[F(\hat{x}) - F(x^\star)] \le \delta$.

**Hardness in our setting.**    What is the minimum number of gradient evaluations required to sample from a target distribution satisfying Assumptions 1–3 with fixed TV error $\varepsilon > 0$, given only access to the gradients $\nabla f_k$, $0 \le k \le T$? In this section we show (informally) by counterexample that one needs to compute at least $\bar{\Omega}(T)$ gradients to sample with TV error $\varepsilon \le \frac{1}{20}$. As a counterexample, consider the Bayesian linear regression posterior considered in Section G, with $d = 1$. Suppose that one only computes stochastic gradients using gradients with index in a random set $S_i = \{\tau_1, \dots, \tau_{\frac{T}{2}}\}$, of size $\frac{T}{2}$, where each element of $S_i$ is chosen independently from the uniform distribution on $\{1, \dots, T\}$. Then the mean of these stochastic gradients (conditioned on the subset $S_i$) are gradients of a function $-\log(\hat{\pi}^{(i)})$, for which $\hat{\pi}^{(i)}$ is the density of the normal distribution $N(\mu_i, \frac{1}{2t})$, where the mean $\mu_i = \frac{\sum_{k \in S_i} w_k}{t} \sim N(0, \frac{1}{t})$ is itself (conditional on $S_i$) a random variable.

Now consider two independent random subsets $S_1$ and $S_2$ with corresponding distributions $\hat{\pi}^{(1)}$ and $\hat{\pi}^{(2)}$. The means of the distributions $\hat{\pi}^{(1)}$ and $\hat{\pi}^{(2)}$ (conditional on $S_1$ and $S_2$) are independent random variables $\mu_1, \mu_2 \sim N(0, \frac{1}{t})$. Hence, the difference in their means $\mu_1 - \mu_2 \sim N(0, \frac{2}{t})$ is normally distributed with standard deviation $\frac{\sqrt{2}}{\sqrt{t}}$. Thus, with probability at least $\frac{1}{2}$, we have $|\mu_1 - \mu_2| \geq \frac{1}{\sqrt{t}}$. Therefore, since (conditional on $S_1, S_2$) we have $\hat{\pi}^{(i)} \sim N(\mu_i, \frac{1}{2t})$ for $i \in \{1, 2\}$, we must have that $\|\hat{\pi}^{(1)} - \hat{\pi}^{(2)}\|_{\text{TV}} \geq \frac{1}{10}$ whenever $|\mu_1 - \mu_2| \geq \frac{1}{\sqrt{t}}$. That is, $\|\hat{\pi}^{(1)} - \hat{\pi}^{(2)}\|_{\text{TV}} \geq \frac{1}{10}$ occurs with probability at least $\frac{1}{2}$. Therefore, one cannot hope to sample from $\pi_T$ with TV error $\varepsilon < \frac{1}{20}$ by using the information from only $\frac{T}{2}$ gradients. One therefore needs to compute at least $\Omega(T)$ gradients to sample from $\pi_T$ with TV error $\varepsilon < \frac{1}{20}$.

# I   Miscellaneous inequalities

We give some inequalities used in the proofs in Section E.

**Lemma I.1.** *Suppose that $X_t$ are a sequence of random variables in $\mathbb{R}^d$ and for each $t$, $\|X_t - \mathbb{E}[X_t|X_{1:t-1}]\|_\infty \leq M$ (with probability 1). Let $S_T = \sum_{t=1}^T \mathbb{E}[X_t|X_{1:t-1}]$ (a random variable depending on $X_{1:T}$). Then*

$$\mathbb{P}\left(\left\|\sum_{t=1}^T X_t - S_t\right\|_2 \geq c\right) \leq 2de^{-\frac{c^2 T}{2M^2 d}}. \tag{186}$$

*Proof.* By Azuma's inequality, for each $1 \leq j \leq d$,

$$\mathbb{P}\left(\left|\sum_{t=1}^T (X_t)_j - (S_t)_j\right| \geq c\right) \leq 2e^{-\frac{c^2 T}{2M^2}}. \tag{187}$$

By a union bound,

$$\mathbb{P}\left(\left\|\sum_{t=1}^T X_t - S_t\right\|_2 \geq c\right) \leq \sum_{j=1}^d \mathbb{P}\left(\left|\sum_{t=1}^T (X_t)_j - (S_t)_j\right| \geq \frac{c}{\sqrt{d}}\right) \leq 2de^{-\frac{c^2 T}{2M^2 d}}. \tag{188}$$

$\square$

**Lemma I.2.** *Suppose that $\pi$ is a distribution with $\mathbb{P}_{\theta \sim \pi}(\|\theta - \theta_0\| \geq \gamma) \leq Ae^{-k\gamma}$, for some $\theta_0$. Then*

$$\mathbb{E}_{\theta \sim \pi}[\|\theta - \theta_0\|^2] \leq \left(2 + \frac{1}{k}\right)\log\left(\frac{A}{k^2}\right).$$

*Proof.* Without loss of generality, $\theta_0 = 0$. Then

$$\mathbb{E}_{\theta \sim \pi}[\|\theta\|^2] = \int_0^\infty 2\gamma \mathbb{P}_{\theta \sim \pi}(\|\theta\| \geq \gamma)\, d\gamma \tag{189}$$

$$\leq \gamma_0 + \int_{\gamma_0}^\infty 2\gamma \mathbb{P}_{\theta \sim \pi}(\|\theta\| \geq \gamma)\, d\gamma \tag{190}$$

$$\leq \gamma_0 + \int_{\gamma_0}^\infty 2\gamma Ae^{-k\gamma}\, d\gamma \qquad \text{by assumption} \tag{191}$$

$$= \gamma_0 + A\left(-\frac{2\gamma}{k}e^{-k\gamma}\Big|_{\gamma_0}^\infty - \int_{\gamma_0}^\infty -\frac{2}{k}e^{-k\gamma}\, d\gamma\right) \qquad \text{integration by parts} \tag{192}$$

$$= A\left(\frac{2\gamma_0}{k}e^{-k\gamma_0} + \frac{2}{k^2}e^{-k\gamma_0}\right). \tag{193}$$

Set $\gamma_0 = \frac{\log\left(\frac{A}{k^2}\right)}{k}$. Then this is $\leq \left(2 + \frac{1}{k}\right)\log\left(\frac{A}{k^2}\right)$, as desired. $\square$