[Reviews · NeurIPS 2019]

Reviewer 1



EDIT: Thanks to the authors for addressing my questions in the rebuttal. I have read the other reviews and will still recommend acceptance. I want to emphasize that addressing the clarity concerns of all three reviewers will improve the impact of an already strong result. Quality: The paper is of high quality and the proofs seem mostly correct. A few comments: - I would suggest that the authors consider putting Figure 1 into the main body of the paper. It will help give readers a sense of the gains that are practically possible. - Eq (14) of the Appendix, should this be eta instead of eta_t? - Eq (22) of the Appendix, x_i should be capitalized, I believe. - Line 108, "f_t's are iid", I'm not sure what this means since f_t are not random variables. Originality: The algorithm itself is primarily an application of SAGA as an inner loop, but the analysis is original and the observation that the constants can be balanced in an appropriate way is original. Significance: As mentioned in the contributions section, this contribution is significant for two reasons. (1) the algorithm may be used in practice, assuming the hyperparameters can be reasonably tuned, and (2) the assumptions and proofs are of independent interest. - My only concern with the current results is that the only example given is Bayesian logistic regression, for which a specialized method already exists. It would be nice if the authors could comment on other possible scenarios. Clarity: The draft could use some work with the presentation. While the writing and notation are generally clear, I think the authors could make a few changes that might make the paper easier to read for a broader audience. - A great deal of space is taken to discuss how this algorithm and the results compare to the complexity of other results in the literature. Obviously this is a central contribution and should be given good real estate, but I think the authors could condense this discussion to a single section, instead of distributing it across multiple sections. This would free up space to discuss the intuitive content of the results and algorithms. - I would spend a bit more time in the introduction explaining in more intuitive terms the information that this algorithm exploits and why that information is sufficient to recover these results. This should give readers a clearer sense how the assumptions prevent non-degeneracy without strong convexity.

Reviewer 2



EDIT: I have read the rebuttal and the other reviews, and I've increased my score to 6 as I'm reasonably confident that the authors will satisfyingly reorganise the paper. # Summary of the paper The paper is a theoretical investigation of the problem of online sampling of log-concave distributions. The authors propose an online variant of stochastic gradient Langevin descent with variance reduction. In the case of Bayesian inference with T data items, the resulting algorithm can reach a small total variation ball around the target distribution in time T log(T). # Summary of the review Overall, the content is very interesting and the results seem new and potentially impactful. Looking at sampling with a fresh optimization eye brings new ideas. However, this paper is clearly more fit for a journal than for a conference like NIPS: all results are shortened versions of results in the 30-page supplementary, and all vertical spaces have been suppressed in the main 8-pager to fit in as much text as possible. The resulting main paper is extremely dense, with important basic examples reduced to a minimum, while not providing enough detail for a stand-alone read. Beyond the ill-formatted paper, the major contributions are subtle and technical, like the exit time argument for martingales. The current reviewing time allocated for conference papers is so short that I cannot carefully read the long appendix, which contains the important proofs and definitions. To sum up, I am in a difficult spot to make a decision, because the contribution seems both interesting and substantial, but I do not think NIPS is the right outlet -in its current form- for such papers, both in terms of format and reviewing mode. More contentwise, can the authors comment on my points 1 and 2 below? # General comments 1. The main motivation seems to be Bayesian inference. But in this case, the user's objective is to integrate functions. Showing that one can reach a small TV ball is only an intermediate step, and the useful result is usually a CLT that guarantees that we can use the history of the Markov chain to estimate integrals. Do you have a CLT for your chain? Does it have \sqrt{T} rate? What makes me prompt the question is that SGLD, for instance, has been shown to lead to a CLT with a slower rate than vanilla Metropolis-adjusted Langevin [Thiery, Teh, Vollmer, Arxiv:1409.0578]. 2. To play the devil's advocate, it seems that your algorithm will perform well only when the posterior is sufficiently concentrated around the MAP (e.g. L210). In that case, wouldn't a simple Laplace approximation be as good in practice? You say L323 that not all posteriors can be approximated by a Gaussian. I agree, but it is not obvious to me that there are interesting practical problems with a non-Gaussian posterior, and still enough concentration that your Assumptions would hold. Maybe a simple experimental comparison would be enough to clear my doubts. Since Bayesian logistic regression is a main motivation, a simple experiment which compares your algorithm with Laplace and, say, SGLD would be a good addition, even with toy data. For such a theoretical paper, a rigorous experiment is also a useful sanity check. 3. The text is very dense, both in content and in shape. You have artificially reduced all vertical spaces to the minimum, which gives a bad visual impression (look at p7 with all the maths in-line and many references to the appendix that force the reader to go back and forth). I am genuinely interested in the results, but my eyes and brain are reluctant to delve into the massive block of text.

Reviewer 3



------------------------------------------------------ After rebuttal: Thank you for the responses. As all the reviewers agreed, the paper is interesting and strong and definitely needs a better organization. In that sense, the authors should consider all the suggestions including moving Sec 2.3 to the appendix and mentioning it from the main text. On the other hand, as this is a single-cycle reviewing process, I will keep my current grade as is, which I believe is fair for this submission in its current form (without the potential improvements, which won't go through reviewing). ------------------------------------------------------ Originality: The overall method is basically a modification of the variance-reduced SGLD algorithm on an online sampling problem. While the method is not very original, online sampling with SG-MCMC algorithms has been an open problem and the whole paper (i.e. the way the algorithm is applied on online learning + theoretical guarantees + the proof technique) is sufficiently original. Quality: The paper is quite solid in terms of theoretical rigor. The authors summarize their proof technique in Section 4, which I believe establishes the main theoretical contribution of the paper. I didn’t go over the proofs line by line but they look correct overall. I also particularly enjoyed section 2.1, which provides important intuition about the assumptions. The experiments are rather weak; however, as the paper is a theoretical paper, I think they are sufficient. Clarity: My main concern is about clarity. Even though the grammar and the language are quite good, I must admit the organization of the paper is rather poor. The authors explain the theoretical results before they explain the algorithm. There several algorithm-specific terms that appear in the theory and they are not even properly introduced before page 6. Before reading the whole paper, it’s not clear why section 2.3 exists. In its current version, it is unnecessarily difficult to read/follow the paper. I have decided to lower my overall grade due to this issue, which I believe is an important one, given that the papers are written in order to communicate information. Some small points: At the beginning, it’s not clear why t should start from 1. This becomes clear when the authors state f0 is the prior and f1 is the first likelihood term. This should be made clear in problem 1.1, otherwise it’s a little bit mysterious. The title of section 3 is misleading. It doesn’t really explain the proof techniques. Significance: Online sampling with SG-MCMC algorithms haven’t been investigated in detail. In that respect, the paper addresses an important problem and is definitely of interest to the community.

[Author Response · NeurIPS 2019]

**Reviewer #1:** Thank you for the insightful comments. We are encouraged you find our paper a significant contribution.

*"many hyperparameters..."* Our results hold even if the parameters $i_{\max}, c, b$ are set above the values suggested, or if the
parameters $\eta_0, C'$ are set below the values suggested in Theorem A.7. In the simulations, we found a fixed batch size of
$b = 64$ and a step size of $\eta_t = \frac{\eta_0}{c+t}$ (without the resetting step) attains a marginal accuracy equal to that attained by the
Pólya-Gamma method specialized to logistic regression. In this case, there are just two hyperparameters $(\eta_0, c)$ to tune.

*"$f_t$ are not random variables"* In some applications, one observes iid random variables and defines functions $f_t$ based on
them, which are then iid random functions. For example, for logistic regression, an iid observation $(u_t, y_t)$ leads to iid
random functions $f_t(\theta) = -\log(1/(1 + e^{-y_t u_t^\top \theta}))$ (lines 179–186). However, our main results (Theorems 2.1 and 2.2)
do not require the $f_t$'s to be iid random.

*"specialized method already exists"* The specialized method (Pólya-Gamma sampler) has running time at each epoch that
scales linearly with $t$, while our algorithm scales as $\mathrm{polylog}(t)$. Moreover, it is unknown if it attains TV-error $\varepsilon$ in time
polynomial in $\frac{1}{\varepsilon}, t, d$, and other problem parameters.

*"authors could comment on other possible scenarios"* We focused on logistic regression as it is one of the most common
applications. Our method also applies to other log-concave distributions (exponential, Laplace, Dirichlet, gamma, beta,
chi-squared, etc.). Another potential application is for online inference for Gaussian processes (for example, an online
version of [Filippone and Engler, ICML 2015]); we will add a discussion on this to the paper.

**Reviewer #2:** Thank you for the valuable feedback. We are encouraged you find our paper interesting and substantial.

*"main motivation ... estimate integrals"* We agree that Bayesian inference is an important motivation of our work.
However, there are also interesting applications that make use of sampling but not integration, such as reinforcement
learning (via Thompson sampling), or online optimization. We describe these applications in lines 44–50 of our paper.

*"Do you have a CLT for your chain?"* We do not have a CLT for our chain, but believe it is possible to use our results to
prove a CLT. We would do this by modifying our Lemma A.3 to show that the samples generated by our algorithm stay
within a Euclidean ball of radius $r > 0$ with probability roughly $1 - e^{-r}$, and then applying our bound for convergence
in TV distance. The rate would not be $\sqrt{i}$ (we use $i$ to represent the number of Markov chain steps). Although MALA
can attain a $\sqrt{i}$ rate, it has the serious drawback in the online setting that the number of gradients to compute at each
step scales linearly with the number of component functions $t$, while our algorithm scales poly-logarithmically with $t$.

*"experimental comparison..."* At your suggestion, we performed initial experiments comparing the full Laplace approxi-
mation to SAGA-LD on online logistic regression, and found they attain marginal accuracy within $0.003$ of each other.
We note that the full Laplace approximation currently requires optimizing a sum of $t$ functions, which has runtime that
scales linearly with $t$ at each epoch, while our method only scales as $\mathrm{polylog}(t)$. It is an interesting open problem to
compute the Laplace approximation with runtime scaling as $\mathrm{polylog}(t)$. In previous experiments, we found the marginal
accuracy for our SAGA-LD algorithm to be $0.921$, for *online* Laplace to be $0.571$, and for SGLD to be $0.442$ (Figure 1
in the appendix). As part of our reorganization, we will include a section on experiments in the main body of the paper.

*"genuinely interested in the results... massive block of text"* We are very sorry we caused you discomfort in reading the
paper. We will ensure the final version is self-contained and friendly to the reader. Specifically, we will (1) move the
offline sampling results (Section 2.3) to the appendix, (2) streamline the related work section, (3) shorten the proof
overview, and (4) use the freed-up space to include more examples and intuition, and give full statements of theorems.

*"contribution seems both interesting and substantial, but I do not think NIPS is the right outlet."* We are glad that you
find the result interesting, and agree that the format of the paper should be improved for a NeurIPS audience. We will
do this using the steps above, and welcome any other suggestions you may have. We hope you will consider increasing
your score and supporting the paper.

**Reviewer #3:** Thank you for the helpful suggestions to improve the flow of the paper, and for the encouraging review.

*"not clear why Section 2.3 exists"* The offline sampling problem is well studied by the ML community. (See e.g.
[DRW+16, CFM+18] in our paper's references.) We show our algorithm can succeed in this setting under weaker
conditions than in the literature, for a class of weakly convex log-densities under a cold start $X_0$. If the reviewers find it
less interesting than the online problem, we will move Section 2.3 to the appendix to better focus on the online problem.

*"not clear why t should start from 1"* We will explain that $f_0$ can be thought of as a prior before stating Problem 1.1.

*"title of Section 3 is misleading"* We will change the title of Section 3 to "Algorithm for online sampling."

*"The paper should be reorganized to improve clarity..."* We will reorganize the paper's main body in the order you
suggested, shorten the proof overview (moving the more technical parts to the appendix), and make sure all terms are
explained before stating the main results. We hope you will consider increasing your score and supporting the paper.

[Meta-Review · NeurIPS 2019]

All the reviewers and the meta-reviewers are for accepting the paper, given that a decent job is put into rewriting it as indicated in the (updated) reviews. Indeed, the clarity is the main issue with this submission, and we except the authors to address it seriously thanks to the very thorough (all expert) reviews, as they indicate in their response and to their message to the AC. The request of the authors about low score of R2 and the length of the proofs (...) was also taken into the consideration when making the decision.